# TANGO: Clustering with Typicality-Aware Nonlocal Mode-Seeking and Graph-Cut Optimization

Haowen Ma [1]  Zhiguo Long [1]  Hua Meng [2]

## Abstract

Density-based mode-seeking methods generate a *density-ascending dependency* from low-density points towards higher-density neighbors. Current mode-seeking methods identify modes by breaking some dependency connections, but relying heavily on local data characteristics, requiring case-by-case threshold settings or human intervention to be effective for different datasets. To address this issue, we introduce a novel concept called *typicality*, by exploring the *locally defined* dependency from a *global* perspective, to quantify how confident a point would be a mode. We devise an algorithm that effectively and efficiently identifies modes with the help of the global-view typicality. To implement and validate our idea, we design a clustering method called TANGO, which not only leverages typicality to detect modes, but also utilizes graph-cut with an improved *path-based similarity* to aggregate data into the final clusters. Moreover, this paper also provides some theoretical analysis on the proposed algorithm. Experimental results on several synthetic and extensive real-world datasets demonstrate the effectiveness and superiority of TANGO. The code is available at `https://github.com/SWJTU-ML/TANGO_code`.

## 1. Introduction

Density-based clustering methods (Hartigan, 1975; Ester et al., 1996; Tobin & Zhang, 2024) have gained wide attention and thorough investigation due to their capability in handling complex data distributions. Mean Shift (Cheng, 1995)

[1]School of Computing and Artificial Intelligence, Southwest Jiaotong University, Chengdu, China [2]School of Mathematics, Southwest Jiaotong University, Chengdu, China. Correspondence to: Zhiguo Long <zhiguolong@swjtu.edu.cn>, Hua Meng <menghua@swjtu.edu.cn>.

*Proceedings of the 42nd International Conference on Machine Learning*, Vancouver, Canada. PMLR 267, 2025. Copyright 2025 by the author(s).

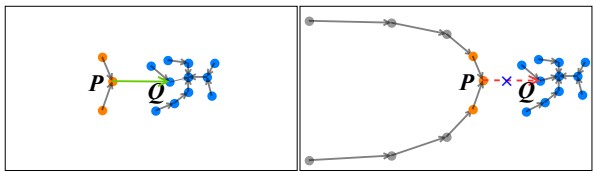

*Figure 1.* Illustration of that whether point $P$ should be considered as a mode is influenced by the global structure of the dataset, which varies with the presence of gray points. Here, each data point links to its nearest neighbor of higher density.

and Quick Shift (Vedaldi & Soatto, 2008) are two of the most representative density-based mode-seeking methods. Mean Shift iteratively converges data points into multiple clusters along the path of the steepest ascent of the density function. In contrast, Quick Shift enhances efficiency by establishing a *density-ascending dependency* through directly linking each data point to its nearest neighbor with higher density and uses a distance threshold to break the dependency to obtain modes and clusters, eliminating the iterative process of Mean Shift. However, Quick Shift is highly sensitive to the threshold parameter, and it always misidentifies outliers as modes.

Building upon this, Density Peaks Clustering (DPC) (Rodriguez & Laio, 2014) introduces a novel approach to detect modes, where modes of the data (cluster centers) are selected based on a decision graph given by both the local density and the distance to the nearest point with higher density. Each non-center point continues following the density-ascending dependency like Quick Shift, ultimately connecting to a mode (cluster center) to complete clustering. Subsequent researchers have proposed various improvements of DPC tailored for manifold-structured, unevenly dense, and noisy data (Liu et al., 2018; Wang et al., 2023).

However, local data characteristics such as density and distance of neighboring points are sometimes not enough to appropriately determine if a data point should be considered as a mode. As illustrated in Fig. 1 (left), when the dataset consists of only the blue and orange points, the point $P$ represents the density peak of a sparse cluster containing only a few points. In this scenario, such a small cluster would usually be considered as outliers and should be merged into a

neighboring higher-density cluster, indicating that $P$ should not be identified as a mode in this case.

On the other hand, when the gray points are included in the dataset (right, Fig. 1), the cluster whose density peak is $P$ becomes substantially larger and forms a distinct group within the data distribution. In this case, merging it into another cluster would be inappropriate, suggesting that $P$ should indeed be recognized as a mode.

In fact, despite the local distribution around $P$ remains constant, the varying global patterns will change the *confidence* for $P$ to be a mode. Existing approaches like (Vedaldi & Soatto, 2008; Rodriguez & Laio, 2014; Jiang et al., 2018; Wang et al., 2023) can only address such scenarios through case-by-case threshold settings or substantial human intervention, which is ineffective across datasets or even impractical.

We found that the confidence for a point $P$ to be a mode can be naturally revealed from a global perspective via the density-ascending dependency. Specifically, we introduce the concept of *typicality*, which measures how confident a point would be a mode, by considering the collective set of points that ultimately converge toward it through the density-ascending dependency, as well as the strength of the dependencies. Points with higher typicality are more likely to be modes, and by comparing the typicality of a point with that of its higher-density neighbor, we can determine whether to break the dependency at this point to form a mode. For example, in Fig. 1 (left), the typicality of $P$ is lower than that of $Q$, and the dependency from $P$ to $Q$ should be retained; while in Fig. 1 (right), the typicality of $P$ is higher than that of $Q$ and the dependency should be broken. In this way, typicality enables mode-seeking that eliminates the need for more human intervention in such cases and better captures the inherent cluster structure.

In this work, we develop a novel clustering algorithm *TANGO*, which leverages the proposed typicality to detect modes and employ graph-cut technique (Ng et al., 2001; Long et al., 2022) with an improved path-based similarity to aggregate data into final partition. Particularly, we first partition the data into tree-like sub-clusters by a typicality-aware mode-seeking approach; then, the inter sub-cluster similarities are evaluated through an improved path-based connectivity (Fischer et al., 2003; Chang & Yeung, 2008; Little et al., 2020); and finally, by using a graph-cut method, we aggregate the sub-clusters from a global perspective.

The contributions of this paper are as follows:

- By integrating local and overall distribution characteristics of data, a new clustering framework that fuses local and global information is proposed, introducing a new global perspective into mode-seeking methods.

- We introduce typicality to reveal the global importance of data points under locally defined density-based dependency, and utilize it to detect modes in a way without human intervention; we also provided theoretical analysis on the uniqueness and statistical property of typicality, as well as on computational efficiency of the algorithm.

- An improved path-based similarity is devised to comprehensively and efficiently evaluate the similarity of sub-clusters and graph-cut is employed to determine the final clusters.

The rest of this paper is organized as follows: The next two sections review related work and preliminaries, and then we present the details of the proposed TANGO algorithm, followed by a comparison of TANGO with multiple state-of-the-art algorithms on extensive datasets through experiments.

## 2. Related Work

Our proposed method leverages mode-seeking and spectral clustering. The following discussion reviews the corresponding relevant literature.

### 2.1. Density-Based Mode-Seeking

Density-based clustering by mode-seeking relies on density estimation, establishment of dependency relationships and identification of modes (e.g., density peaks).

The earlier Mean Shift (Cheng, 1995) iteratively moves each data point to the mean of its neighborhood, which is equivalent to establishing dependency relationships in the direction of the steepest ascent of density (Arias-Castro et al., 2016). Quick Shift (Vedaldi & Soatto, 2008) establishes dependency relationships by directly associating each point with its nearest neighbor of higher density, and identifies *modes* as points not depending on others, which significantly accelerates Mean Shift. HQuickShift (Altinigneli et al., 2020) integrates hierarchical clustering with Quick Shift and a Mode Attraction Graph to help determine the hyperparameter for establishing dependencies, to deal with the sensitivity to hyperparameters of Quick Shift. Quick Shift++ (Jiang et al., 2018) improves Quick Shift by computing density using k-nearest neighbors and identifying modal-sets (Jiang & Kpotufe, 2017) instead of point as modes, which are connected components of mutual neighborhood graph determined by a density fluctuation parameter. DCF (Tobin & Zhang, 2021) improves the efficiency of modal-sets identification based on Quick Shift++. CPF (Tobin & Zhang, 2024) reduces the sensitivity to the density fluctuation parameter.

Compared with Quick Shift and its variants, the essential difference of DPC (Rodriguez & Laio, 2014) is that it iden-

tifies modes by a decision graph of density and distance. Optimizations of DPC usually focus on density estimation, modes identification, and dependency relationships. DPC-DBFN (Lotfi et al., 2020) employs fuzzy kernels to compute density and identifies connected components containing cluster centroids in mutual KNN graphs. SNN-DPC (Liu et al., 2018) considers shared nearest neighbors to better characterize similarities and further refines the dependency relationships for boundary points by a KNN voting strategy. VDPC (Wang et al., 2023) utilizes two user-defined parameters to optimize dependency relationships. DPC-GD (Du et al., 2018) and DLORE-DP (Cheng et al., 2020) address manifold-structured data by substituting Euclidean distance with geodesic distance. However, these approaches only focus on locally defined density and dependency, lacking global characteristics of data.

There also exist methods that initially create sub-clusters by utilizing the density-ascending dependency derived from DPC and Quick Shift, and then aggregates them using other clustering techniques to obtain final clustering results. For example, to aggregate sub-clusters, FHC-LDP (Guan et al., 2021) and LDP-MST (Cheng et al., 2021) use hierarchical clustering; LDP-SC (Long et al., 2022) and DCDP-ASC (Cheng et al., 2022) use spectral clustering; NDP-Kmeans (Cheng et al., 2023) applies K-means clustering with geodesic distance; DEMOS (Guan et al., 2023) uses a linkage-based approach to aggregate these sub-clusters.

## 2.2. Spectral Clustering

Spectral clustering algorithm (Ng et al., 2001; von Luxburg, 2007) is a classical graph-cut method, which partitions the vertices of a graph by applying K-means on the spectral embeddings obtained through the eigenvectors of the graph Laplacian matrix. It was noticed by (Lee et al., 2014) that the $k$-th smallest eigenvalue of the normalized Laplacian matrix determines whether a graph could be well-partitioned into $k$ groups, and the upper bound on the number of vertices misclassified by spectral clustering has been proven by (Kolev & Mehlhorn, 2016; Macgregor & Sun, 2022; Mizutani, 2021; Peng et al., 2017). To improve the efficiency of spectral clustering, sampling-based methods (Yan et al., 2009; Chen et al., 2011; Cai & Chen, 2015; Huang et al., 2020) were used to reduce the size of similarity matrix and the cost of eigen-decomposition, and there is also some method (Macgregor, 2023) that leverages a power method to completely avoid the time-consuming eigen-decomposition. When the number of target clusters is not pre-specified in spectral clustering, a series of methods (Xiang & Gong, 2008; von Luxburg, 2007; Brandes et al., 2008; Zelnik-Manor & Perona, 2004) have been developed to deal with this situation.

---

**Algorithm 1** Quick Shift

**Require:** Dataset $X$, similarity matrix $A$, density $\rho$, similarity threshold $\tau$
**Ensure:** Mode set mode, Clustering results labels
 1: mode $\leftarrow \varnothing$; Initialize dependency matrix $B$ as a zero matrix
 2: **for** each $x_i \in X$ **do**
 3:     $x_j \leftarrow \underset{x_j : \rho_j > \rho_i}{\mathrm{argmax}}\, A_{ij}$
 4:     $B_{ij} \leftarrow A_{ij}$
 5: **for** each point $x_i$ with $B_{ij} \neq 0$ **do**
 6:     **if** $B_{ij} < \tau$ **then**
 7:         $B_{ij} \leftarrow 0$
 8:         mode $\leftarrow$ mode $\cup \{x_i\}$
 9: Initialize cluster label array labels
10: **for** each $x_i \in X$ **do**
11:     $x_c \leftarrow x_i$
12:     **while** $x_c \notin$ mode **do**
13:         Find $j$ where $B_{cj} \neq 0$
14:         $x_c \leftarrow x_j$
15:     labels$(x_i) \leftarrow$ labels$(x_c)$
16: **return** mode, labels

---

## 3. Preliminaries

Let $X = \{x_1, \ldots, x_n\}$ denote a set containing $n$ data points, where $x_i \in \mathbb{R}^d$ represents a data point. Let $A \in \mathbb{R}^{n \times n}$ be the similarity matrix among the data points, and $\rho_i$ denote the density of $x_i$. Assuming that a dependency matrix $B \in \mathbb{R}^{n \times n}$ has been constructed based on $A$, where $B_{ij}$ represents the dependency from $x_i$ to $x_j$. The specific definitions of these concepts will follow later in Section 4. In this section, we briefly introduce how Quick Shift and DPC work.

### 3.1. Quick Shift

As described in Algorithm 1, for each data point $x_i \in X$, Quick Shift sets $B_{ij} = A_{ij}$ if and only if $x_j$ is the nearest higher-density neighbor of $x_i$, to establish the density-ascending dependency. Quick Shift then breaks the dependency at point $x_i$ and recognize $x_i$ as a mode if $A_{ij}$ is less than the human-determined similarity threshold. All points that converge to the same mode through the dependency are considered as in the same cluster.

### 3.2. Density Peaks Clustering

The only difference between Density Peaks Clustering (DPC) and Quick Shift is that DPC considers not only similarity but also density to break the dependency and obtain modes. Specifically, DPC breaks the dependency from $x_i$ to its nearest higher-density neighbor $x_j$ if $A_{ij}$ is less than the similarity threshold and $\rho_i$ is also larger than the den-

sity threshold simultaneously, avoiding identifying some outliers as modes.

In fact, not only Quick Shift and DPC, but also many other existing mode-seeking approaches (Jiang et al., 2018; Wang et al., 2023; Guan et al., 2023) require human intervention or case-by-case threshold setting to identify modes effectively.

## 4. The Proposed TANGO Algorithm

The overall framework of the proposed TANGO algorithm is shown in Figure 2. TANGO will first construct a density-ascending dependency graph by exploring local similarity and density structures; then evaluate the typicality of points by exploring the dependency from a global perspective, and break some of the dependency connections by exploiting typicality to identify modes and sub-clusters; and finally, use a graph-cut with an improved path-based similarity to aggregate sub-clusters to achieve clustering.

### 4.1. General Definition for Typicality

In the traditional mode-seeking framework, algorithms like Quick Shift and DPC make data points depend on their nearest higher-density neighbors to generate a density-ascending dependency. While these methods balance efficiency and accuracy, with some theoretical guarantees (Jiang, 2017; Tobin & Zhang, 2024), they primarily consider local information such as density and similarity to break the dependency and obtain modes, making them require extensive human intervention such as case-by-case threshold setting or manual operation to identify modes effectively.

To address this drawback, we propose to further explore the dependency relationship to model a global-view "typicality" of points to quantify the confidence for them to be modes. We suppose that typicality should satisfy the following postulates:

- **(P1)** Typicality should consider the own density of a point. Points with higher density may have higher confidence to be modes, as they represent significant concentrations in the data distribution.

- **(P2)** The typicality of a point should also be related to all points that ultimately converge toward it through the density-ascending dependency, as well as the strength of the dependencies (matrix $B$). If a point accumulates stronger dependencies from its descendant points, then this point is more likely to be a genuine mode.

Given these postulates, we can define typicality $T_i$ for each point $x_i$ generally as:

$$T_i = \rho_i + \sum_j B_{ji} T_j \tag{1}$$

where $B_{ji} \neq 0$ iff there is a dependency from $x_j$ to $x_i$. Note that the definition of typicality in Eq. (1) is recursive and actually captures the weighted influence of all points that converge to $x_i$ through the dependency relationship. For instance, as shown on the right side of Figure 2, we have $T(x_i) = B_{1i}T(x_1) + B_{2i}T(x_2) + B_{3i}T(x_3) + \rho_i$, and $T(x_1) = \rho_1$, $T(x_2) = B_{42}T(x_4) + \rho_2$, $T(x_3) = \rho_3$, $T(x_4) = \rho_4$. Then $T(x_i) = B_{1i}\rho_1 + B_{2i}(B_{42}\rho_4 + \rho_2) + B_{3i}\rho_3 + \rho_i$, indicating that $x_i$ collects typicality from $x_1$, $x_2$, $x_3$ and $x_4$, which are all points that converge to it.

Let $T = (T_1, T_2, ..., T_n)^\intercal$ and $\rho = (\rho_1, \rho_2, ..., \rho_n)^\intercal$ be two column vectors. Eq. (1) can be rewritten as:

$$T = B^\intercal T + \rho. \tag{2}$$

**Remark 1.** If we consider each data point in a dataset as an individual webpage weighted by its density, and for each page there exist hyperlinks linking to its neighboring higher-density pages. Thus, the proposed typicality can also be considered as a special kind of *PageRank Centrality* (Brin & Page, 1998; Avrachenkov & Lebedev, 2007; Avrachenkov et al., 2018; Chen et al., 2017; Litvak et al., 2006; Volkovich et al., 2007), where $B_{ij}$ denotes the probability from $x_i$ jumping to $x_j$. When $x_i$ is a distinct mode, it creates a so-called "attraction basin" – a region where all points ultimately converge toward $x_i$ through the dependency, and $x_i$ collects typicality from all points in its "basin".

### 4.2. Typicality Based on Hierarchical Dependency

We propose a specific typicality measure based on hierarchical dependency, where each data point only depends on its nearest neighbor with higher density. Hierarchical dependency has been widely used in many density-based clustering methods such as (Vedaldi & Soatto, 2008; Rodriguez & Laio, 2014; Jiang et al., 2018; Guan et al., 2023; Tobin & Zhang, 2024; Cheng et al., 2021). Its consistency guarantee has also been theoretically proven by several articles such as (Jiang, 2017; Tobin & Zhang, 2024). First, we introduce some basic definitions to uncover such dependency within the dataset. Subsequently, we analyze the theoretical properties of corresponding typicality.

Complex data often exhibit specific manifold structures and its similarity can be better reflected by shared nearest neighbor information (Jarvis & Patrick, 1973; Liu et al., 2018). Therefore, we define a similarity measure based on shared nearest neighbors, and we further distinguish the different contribution to similarity of each shared neighbor to have better robustness.

**Definition 1** (Similarity). For any two data points $x_i, x_j \in X$, let $N_k(x_i)$ and $N_k(x_j)$ denote the $k$ nearest neighbors of $x_i$ and $x_j$, respectively, and $\mathsf{SNN}_k(x_i, x_j) = N_k(x_i) \cap N_k(x_j)$. The *similarity* $A(x_i, x_j)$ (or $A_{ij}$) is defined as: $A(x_i, x_j) = \sum_{p \in \mathsf{SNN}_k(x_i, x_j)} \exp(-(\frac{d(p, x_i) + d(p, x_j)}{2d_{\max}})^2)$ if

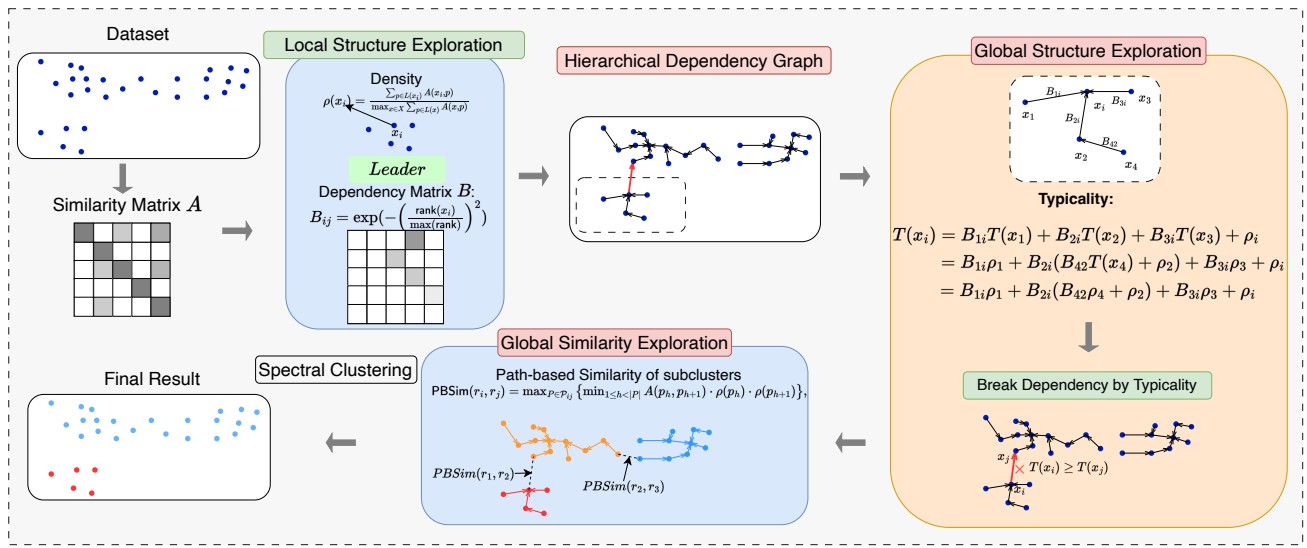

*Figure 2.* The framework of the proposed TANGO algorithm.

$x_i \in N_k(x_j)$ or $x_j \in N_k(x_i)$, and 0 otherwise. Here, $d$ denotes the Euclidean distance between two points, and $d_{\max}$ is the maximum Euclidean distance between any point and its $k$ nearest neighbors in the dataset.

Based on the above definition of similarity, we define the *density* of points as usual with normalization.

**Definition 2** (Density). For $x_i \in X$, let $L(x_i)$ denote the set of $k$ points most similar to $x_i$ in terms of the similarity matrix $A$, where $k$ is the same one used in Definition 1. Then, the *density* of $x_i$ is:

$$\rho(x_i) = \frac{\sum_{p \in L(x_i)} A(x_i, p)}{\max_{x \in X} \sum_{p \in L(x)} A(x, p)} \qquad (3)$$

To obtain the dependency matrix $B$, similar to Quick Shift and DPC, we can establish a hierarchical *density-ascending dependency* within the dataset based on similarity and density information as following, known as *leader relationship*.

**Definition 3** (Leader relationship). For $x_i$, let higher$(x_i)$ denote the set of points with higher density than $x_i$ and nonzero similarity to $x_i$, i.e., higher$(x_i) = \{p \in X \mid \rho(p) > \rho(x_i), A(x_i, p) \neq 0\}$. The leader point of $x_i$ is:

$$\mathsf{leader}(x_i) = \begin{cases} \underset{x_j \in \mathsf{higher}(x_i)}{\arg\max} \ A_{ij} & \text{if } \mathsf{higher}(x_i) \neq \emptyset \\ \text{None} & \text{otherwise.} \end{cases} \qquad (4)$$

According to the above definition, $\mathsf{leader}(x_i)$ is the point with higher density than $x_i$ that has the highest similarity to $x_i$. If $x_i$ has zero similarity with all points of higher density, then $x_i$ has no leader. Then the density-ascending

dependency is constructed by allowing each point depending on its leader if it exists.

Beyond that, we should further consider the strength of such dependency. For instance, if a point is closer to its leader, the dependency of them should be stronger. However, directly considering similarity can lead to stronger connections in dense regions and weaker connections in sparse ones. Therefore, we propose considering the position of the leader within the sequence of neighbors sorted by similarity.

**Definition 4** (Rank). For $x_i \in X$, let $N_\downarrow(x_i)$ denote the list of its neighbors sorted in descending order of similarity. Let $x_j = \mathsf{leader}(x_i)$. We define $\mathsf{rank}(x_i)$ as

$$\mathsf{rank}(x_i) = \begin{cases} r & \text{if } x_j \text{ ranks } r\text{-th in } N_\downarrow(x_i) \\ \text{None} & \text{if } x_j = \text{None} \end{cases} \qquad (5)$$

That is, if $x_i$ has a leader, then $\mathsf{rank}(x_i)$ represents the position of $\mathsf{leader}(x_i)$ in the list of neighbors sorted in descending order of similarity to $x_i$.

We then define the *dependency matrix* $B \in \mathbb{R}^{n \times n}$ as follows:

$$B_{ij} = \begin{cases} \exp\left(-\left(\frac{\mathsf{rank}(x_i)}{\max(\mathsf{rank})}\right)^2\right) & \text{if } x_j = \mathsf{leader}(x_i) \\ 0 & \text{otherwise} \end{cases} \qquad (6)$$

where $\max(\mathsf{rank})$ is the maximum value of rank among all points.

Note that since each point has at most one leader, each row of matrix $B$ has at most one non-zero element, and the

dependency relationship is actually in a *hierarchical* form and partitions the data into disjoint tree-like sub-parts.

When computing the typicality $T$, the non-zero elements of matrix $B$ specify the weights of contribution of the points to the typicality of their respective leaders. Using $\mathsf{rank}(x_i)$ to compute the weight can avoid the bias in favor of dense region.

**Theorem 1.** *For $B$ defined in Eq.* (6)*, there exists a unique typicality vector $T$ such that $T = B^\mathsf{T}T + \rho$ (Eq.* (2)*) holds.*

*Proof.* See Appendix A.1. □

As stated in the following theorem, solving for the typicality $T$ defined by Eq. (6) is highly efficient.

**Theorem 2.** *For $B$ defined by Eq.* (6)*, if data points $x_1, \ldots, x_n$ are sorted by density $\rho$ in ascending order, then $T$ satisfying Eq.* (2) *($T = B^\mathsf{T}T + \rho$) can be computed in $O(n)$ time.*

*Proof.* From the proof of Theorem 1, it is known that $I - B^\mathsf{T}$ is a lower triangular matrix with ones on the diagonal when the data points are sorted in ascending order of density. This implies that the system of linear equations $(I - B^\mathsf{T})T = \rho$ can be solved by forward substitution, computing $T_1, T_2, \ldots, T_n$ in sequence. For each data point $x_i$, let $x_{i_1}, x_{i_2}, \ldots, x_{i_{s_i}}$ be the set of points whose leader is $x_i$, we have $T_i = B_{i_1 i}T_{i_1} + \ldots + B_{i_{s_i} i}T_{i_{s_i}} + \rho_i$ ($i_1, \ldots, i_{s_i} < i$ and $\forall t = 1, \ldots, s_i, B_{i_t i} \neq 0$). As the graph induced by $B$ consists of trees, we have $\sum_{i=1}^{n} s_i \leq n - 1$. Thus, computing all $T_i$ requires $O(n)$ operations. □

Based on the above theorem, to calculate $T$, we can first sort $x_1, \ldots, x_n$ by density in *ascending order*, and then compute $T$ sequentially for $i$ from 1 to $n$ as: $T(x_j) \leftarrow T(x_j) + T(x_i) \cdot B_{ij}$ where $x_j = \mathsf{leader}(x_i)$. Note that when $x_i$ contributes typicality to its leader, the typicality of $x_i$ itself ($T(x_i)$) has already been fully determined due to the ascending order of density.

**Remark 2.** Under Theorem 1, we can also directly get the closed form $T = (I - B^\mathsf{T})^{-1}\rho$ and expand it into the summation of infinite series $T = \sum_{l=0}^{\infty}(B^\mathsf{T})^l \rho$, which can be more intuitive and interpretable to understand how typicality comes from, as $(B^\mathsf{T})^l$ captures the weights of all $l$-hop paths in the dependency graph induced by $B$.

### 4.3. Typicality-Aware Mode-Seeking

Typicality considers both local density distributions and global structural information as shown in Eq. (2). Points with higher typicality have higher confidence to be modes. Therefore, we break the dependency with the help of typicality and obtain modes accordingly.

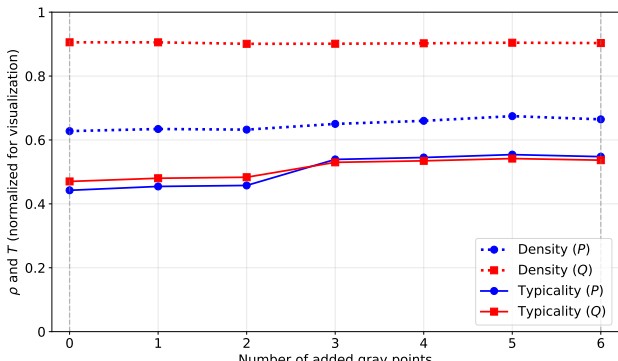

*Figure 3.* Evolution of density and typicality for points $P$ and $Q$ as gray points are incrementally added in Fig. 1.

Specifically, for $x_i$ and its leader $x_j$,

- if $T(x_i) < T(x_j)$, then we keep the dependency from $x_i$ to $x_j$;

- if $T(x_i) \geq T(x_j)$, then we break the dependency and assign $x_i$ as a mode and the root of corresponding tree-like sub-cluster.

**Remark 3.** That is to say, points with higher confidence to be modes should not be represented by those with lower ones. $x_j$ can represent $x_i$ if and only if $x_j$ can accumulate more typicality than $(1 - B_{ij}) \cdot T(x_i)$ from its own density $\rho_j$ and other points excluding $x_i$. Either a larger $T(x_i)$ or a weaker dependency from $x_i$ to $x_j$ would reduce the ability of $x_j$ to represent $x_i$, while the density of $x_j$ and typicality accumulated from other points enhance its ability to represent $x_i$.

Fig. 3 illustrates the evolution of density and typicality for points $P$ and $Q$ as gray points are incrementally added in Fig. 1. The results demonstrate that $T(P)$ keeps growing and ultimately surpasses $T(Q)$, allowing the algorithm to automatically determine whether to break the dependency at $P$ to form a mode. The procedure to calculate typicality and use it to detect modes is given in Algorithm 2.

**Remark 4.** From a stochastic equation perspective, the tail behavior of typicality follows a power law distribution, which means that the possibility of data points having high confidence to be modes decreases polynomially. See Appendix B.

### 4.4. Aggregating Mode-Centered Sub-Clusters

After obtaining the modes and corresponding tree-like sub-clusters based on typicality, we then use graph-cut method to aggregate them into final cluster partitions. To better reflect the inter-sub-cluster similarities based on the overall

---

**Algorithm 2** Typicality-Aware Mode-Seeking

---

**Require:** Dataset $X$; leaders leader; density $\rho$; rank defined in (5); dependency matrix $B$.

**Ensure:** mode.
 1: mode $\leftarrow \varnothing$
 2: **for all** $x_i \in X$ **do**
 3: $\quad T(x_i) \leftarrow \rho(x_i)$
 4: **for all** $x_i \in X$ **sorted by ascending order of** $\rho$ **do**
 5: $\quad x_j \leftarrow$ leader$(x_i)$
 6: $\quad T(x_j) \leftarrow T(x_j) + T(x_i) \cdot B_{ij}$
 7: **for all** $x_i \in X$ **do**
 8: $\quad$ **if** $T(x_i) \geq T($leader$(x_i))$ **or** leader$(x_i) = $ None **then**
 9: $\quad\quad$ mode $\leftarrow$ mode $\cup \{x_i\}$
10: **return** mode

---

data distribution, we introduce an improved *path-based similarity* between them based on the similarity of data points defined in Definition 1 and inspired by (Fischer et al., 2003; Chang & Yeung, 2008; Little et al., 2020). This similarity considers global connectivity in addition to local similarity. Intuitively, the path-based similarity between two sub-clusters is the maximum "connectivity" among all paths connecting them, where the "connectivity" of each path is the minimum similarity between adjacent points on that path.

**Definition 5** (Path-based similarity between sub-clusters). Let $r_i$ and $r_j$ denote the modes of two tree-like sub-clusters $G_i$ and $G_j$ generated by Algorithm 2, and let $G$ be the similarity graph constructed based on the similarity matrix $A$. Define $\mathcal{P}_{ij}$ as the set of all paths in $G$ connecting $r_i$ and $r_j$, and let $p_h$ and $p_{h+1}$ be an adjacent pair of points on a path $P \in \mathcal{P}_{ij}$. The similarity between $G_i$ and $G_j$ is given by the following path-based similarity between their modes:

$$\mathsf{PBSim}(r_i, r_j) = \max_{P \in \mathcal{P}_{ij}} \left\{ \min_{1 \leq h < |P|} C(p_h, p_{h+1}) \right\}, \quad (7)$$

where $C(p_h, p_{h+1}) = 1$ if $\exists G_t$ s.t. $p_h, p_{h+1} \in G_t$, and $C(p_h, p_{h+1}) = A(p_h, p_{h+1}) \cdot \rho(p_h) \cdot \rho(p_{h+1})$ otherwise.

Before computing path-based similarity $\mathsf{PBSim}(r_i, r_j)$, we first adjust all direct similarities by weighting them according to the densities of the points, as $A(p_h, p_{h+1}) \cdot \rho(p_h) \cdot \rho(p_{h+1})$, ensuring that paths through high-density regions have higher "connectivity" and those through low-density regions have lower "connectivity". Furthermore, to ensure that $\mathsf{PBSim}(r_i, r_j)$ is determined by pairs of points belonging to different sub-clusters, we set the similarity between all points in the same sub-cluster to the maximum possible value 1. More formal description about path-based similarity $\mathsf{PBSim}(r_i, r_j)$ can be found in the proof of Theorem 3 in Appendix A.2.

The final clustering result is then obtained by applying spectral clustering on these sub-clusters. Specifically, we consider each mode-centered sub-cluster as a vertex in a similarity graph, where the similarity between these sub-clusters is determined by the above path-based similarity $\mathsf{PBSim}(r_i, r_j)$, and finally the spectral clustering with Normalized Cut (von Luxburg, 2007) is applied to aggregate these vertices into a final partition with the specified number of clusters.

The reason why choosing spectral clustering for merging sub-clusters is that it comprehensively considers a global graph-cut cost of the whole partition, unlike other methods such as hierarchical clustering, which partitions the data greedily and ignores the global impact on the whole partition at each greedy step, thus always achieves an inferior or imbalanced partition.

Algorithm 3 and Fig. 2 outline the overall procedure of the proposed algorithm TANGO. Here, a variant of Kruskal's algorithm is utilized to compute the path-based similarity and the procedure can be highly efficient, as shown in the following theorem.

**Theorem 3.** *Path-based similarity between sub-clusters defined by Definition 5 can be calculated in $O(nk \log(nk))$ time.*

*Proof.* See Appendix A.2. $\qquad\square$

### 4.5. Time Complexity

Let $n, d$ be the number of data points and dimensionality for a dataset. For convenience, we use $q$ to denote the number of modes detected by Algorithm 2. The time complexity of TANGO can be analyzed as follows:

- Line 1: $O(dn \log n + nk^2 d)$ for calculating $A$ with KD-Tree, and $O(kn \log(kn) + kn)$ for computing density and leader.

- Line 2: $O(n \log n)$ for sorting data points and computing $T$ according to Theorem 1.

- Line 3: $O(n)$.

- Lines 4-11: $O(nk \log(nk))$ according to Theorem 3.

- Lines 12-15: $O(q^3 + n)$ for spectral clustering and label assignment.

Therefore, as $k, q \ll n$, the overall time complexity of TANGO is $O(nk^2 d)$, and is dominated by the similarity matrix calculation which can be easily parallelized. The practical running times are shown in Appendix C.4.

**Algorithm 3** TANGO

**Require:** Dataset $X$, parameter $k$.

**Ensure:** Clustering results labels.

1: Calculate $A$ as in Definition 1, $\rho$ as in (3), and leader as in (4)
2: Calculate mode using Algorithm 2
3: edges $\leftarrow \emptyset$, CC$(r_i) \leftarrow i(\forall r_i \in \text{mode})$, PBSim $\leftarrow \mathbf{O}$
4: Prepare edges $\leftarrow \{(C(x_i, x_j), r(x_i), r(x_j)) \mid x_i, x_j \in X \text{ and } C(x_i, x_j) \neq 1\}$ according to Definition 5 where $r(x_i)$ denote the mode of sub-cluster which $x_i$ belongs to.
5: **for all** $(C(x_i, x_j), r(x_i), r(x_j)) \in$ edges **sorted in descending order of** $C(x_i, x_j)$ **do**
6:    **if** CC$(r(x_i)) \neq$ CC$(r(x_j))$ **then**
7:       CC$_i \leftarrow \{p \in \text{mode} \mid$ CC$(p) =$ CC$(r(x_i))\}$
8:       CC$_j \leftarrow \{p \in \text{mode} \mid$ CC$(p) =$ CC$(r(x_j))\}$
9:       CC$(p_i) \leftarrow$ CC$(r(x_j))$ **for all** $p_i \in$ CC$_i$
10:       **for all** $(r_i, r_j) \in$ CC$_i \times$ CC$_j$ **do**
11:          PBSim$(r_i, r_j) \leftarrow C(x_i, x_j)$
12: Apply Spectral Clustering on mode with PBSim to obtain clustering results labels
13: **for all** $x_i \in X \setminus$ mode **do**
14:    $r(x_i) \leftarrow$ the mode of $x_i$
15:    labels$(x_i) \leftarrow$ labels$(r(x_i))$
16: **return** labels

# 5. Experiment

In this section, we first visualize the clustering results of Quick Shift, DPC, and the proposed TANGO algorithm on synthetic datasets; then we compare TANGO with 10 existing clustering methods on 16 real-world datasets, and also apply TANGO to image segmentation tasks to further validate its effectiveness and efficiency; finally, we analyze the effects of the hyperparameter $k$ and proposed components (See detail in Appendix C). We also provide some additional discussion in Appendix C.7.

## 5.1. Visualization on Synthetic Datasets

Fig. 4 shows the visualization of dependency relationships and clustering results for TANGO compared with the classical Quick Shift and DPC on four toy datasets, where the red arrows (with blue cross mark) in the 3rd column indicating the breaking of dependency via typicality, and the 4th column shows the final clustering results for TANGO. It can be observed that TANGO achieves favorable clustering results by detecting modes effectively based on typicality, while Quick Shift and DPC both perform poorer due to only considering the local structure of the data.

We also include a visualization on some highly-noisy synthetic datasets in Appendix C.7.1 to validate the robustness of TANGO to noise.

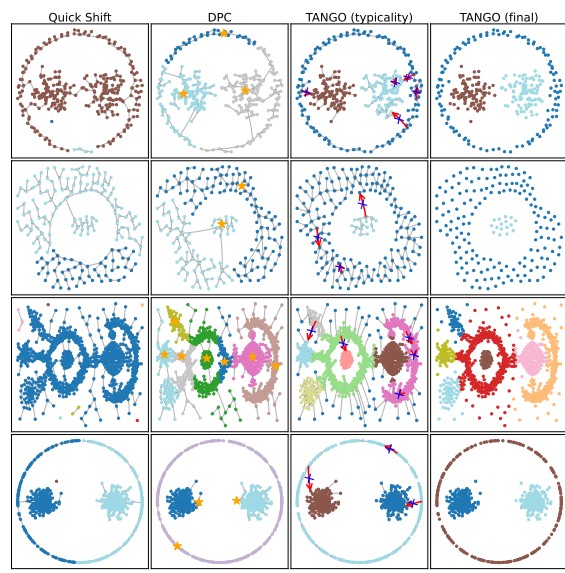

*Figure 4.* Results on 4 synthetic datasets. For TANGO, the hyperparameter $k$ is set to 20, 30, 70, and 30, respectively. For Quick Shift and DPC, we tune their hyperparameters to achieve a reasonable number of clusters while maximizing the Adjusted Rand Index (ARI).

| Dataset | $n$ | $d$ | #Clusters |
|---|---|---|---|
| wdbc | 569 | 30 | 2 |
| heartEW | 270 | 13 | 2 |
| segmentation | 2100 | 19 | 7 |
| semeion | 1593 | 256 | 10 |
| semeionEW | 1593 | 256 | 10 |
| ver2 | 310 | 6 | 3 |
| synthetic-control | 600 | 60 | 6 |
| waveform | 5000 | 21 | 3 |
| CongressEW | 435 | 16 | 2 |
| mfeat-zer | 2000 | 47 | 10 |
| ionosphereEW | 351 | 34 | 2 |
| banknote | 1372 | 4 | 2 |
| isolet1234 | 6238 | 617 | 26 |
| Leukemia | 72 | 3571 | 2 |
| MNIST (AE) | 10000 | 64 | 10 |
| Umist (AE) | 575 | 64 | 20 |

*Table 1.* Summary of real-world datasets.

## 5.2. Clustering on Real-World Datasets

We compared TANGO with 10 advanced algorithms based on density and local dependency relationships, as well as graph-cut clustering, on real-world datasets, including LDP-SC (Long et al., 2022), DCDP-ASC (Cheng et al., 2022), LDP-MST (Cheng et al., 2021), NDP-Kmeans (Cheng et al., 2023), DEMOS (Guan et al., 2023), CPF (Tobin & Zhang, 2024), QKSPP (Jiang et al., 2018), DPC-DBFN (Lotfi et al., 2020), USPEC (Huang et al., 2020), and KNN-SC[1] (tradi-

---

[1]https://scikit-learn.org/stable/index.html

| | ARI | | | | | | | | | | | NMI | | | | | | | | | | | ACC | | | | | | | | | | |
|---|---|---|---|---|---|---|---|---|---|---|---|---|---|---|---|---|---|---|---|---|---|---|---|---|---|---|---|---|---|---|---|---|---|
| | LDP-SC | DCDP-ASC | LDP-MST | NDP-Kmeans | DEMOS | CPF | QKSPP | DPC-DBFN | USPEC | KNN-SC | TANGO | LDP-SC | DCDP-ASC | LDP-MST | NDP-Kmeans | DEMOS | CPF | QKSPP | DPC-DBFN | USPEC | KNN-SC | TANGO | LDP-SC | DCDP-ASC | LDP-MST | NDP-Kmeans | DEMOS | CPF | QKSPP | DPC-DBFN | USPEC | KNN-SC | TANGO |
| wdbc | 76.10 | 0.24 | 28.72 | 44.02 | 49.12 | 43.81 | 52.46 | 79.89 | 79.26 | 81.76 | **85.01** | 65.40 | 0.52 | 31.91 | 43.88 | 47.82 | 39.38 | 42.94 | 69.14 | 68.33 | 73.89 | **77.35** | 93.67 | 62.92 | 77.86 | 83.66 | 85.41 | 80.84 | 76.45 | 94.73 | 94.55 | 95.25 | **96.13** |
| heartEW | 20.80 | 9.67 | -0.19 | 22.90 | 8.03 | 27.09 | 36.62 | 20.07 | 13.37 | 35.76 | **39.40** | 16.26 | 14.13 | 2.04 | 17.72 | 12.72 | 21.24 | 28.87 | 21.57 | 9.54 | 27.95 | **30.70** | 72.96 | 65.93 | 53.33 | 74.07 | 28.89 | 72.96 | 80.37 | 72.59 | 68.52 | 80.00 | **81.48** |
| segmentation | 52.87 | 43.84 | 28.79 | 22.72 | 61.19 | 58.00 | 56.55 | 55.55 | 59.43 | 57.77 | **64.21** | 66.16 | 61.60 | 59.13 | 55.38 | 70.40 | 71.83 | 68.17 | 69.99 | 70.34 | 68.60 | **71.87** | 72.81 | 57.33 | 49.81 | 44.24 | 71.86 | 67.95 | 68.29 | 67.24 | 65.05 | 65.71 | **73.19** |
| semeion | 55.58 | 34.37 | 15.32 | 19.33 | 44.08 | 36.20 | 48.75 | 29.85 | 53.96 | 46.82 | **65.37** | 66.15 | 49.53 | 45.57 | 40.36 | 63.53 | 52.69 | 64.85 | 41.37 | 66.71 | 61.24 | **71.89** | 70.06 | 53.11 | 36.41 | 33.77 | 56.12 | 47.96 | 59.38 | 48.02 | 68.17 | 63.28 | **81.42** |
| semeionEW | 54.92 | 34.15 | 14.01 | 31.04 | 44.14 | 36.04 | 49.05 | 29.69 | 55.95 | 46.53 | **63.49** | 67.33 | 47.36 | 43.49 | 46.21 | 63.54 | 52.14 | 64.17 | 42.04 | 67.64 | 60.49 | **71.11** | 74.07 | 55.24 | 37.41 | 51.98 | 55.43 | 48.02 | 57.75 | 47.96 | 69.43 | 62.90 | **79.35** |
| ver2 | 22.92 | -0.66 | 33.35 | -3.10 | 28.38 | 34.82 | 37.62 | 33.17 | 31.54 | 25.27 | **42.85** | 28.03 | 1.29 | 27.40 | 6.53 | 30.24 | 30.32 | 32.22 | 26.96 | 30.31 | 30.19 | **32.41** | 55.16 | 47.42 | 64.52 | 41.94 | 64.84 | 52.58 | 69.03 | 69.68 | 60.32 | 50.00 | **73.55** |
| synthetic-control | 61.58 | 60.03 | 40.10 | 51.24 | 58.18 | 66.33 | 62.73 | 55.53 | 68.86 | 65.02 | **72.23** | 80.06 | 75.11 | 65.56 | 66.68 | 74.58 | 81.92 | 79.73 | 72.78 | 82.22 | 81.29 | **84.65** | 57.33 | 64.00 | 38.17 | 55.50 | 64.00 | 66.67 | 62.50 | 59.50 | 71.67 | 68.17 | **80.00** |
| waveform | 28.35 | 24.56 | 31.34 | 28.09 | 31.02 | 30.74 | 29.85 | 0.65 | 25.17 | 25.22 | **39.77** | 36.31 | 35.64 | 34.88 | 29.98 | 39.25 | 39.14 | 36.84 | 5.63 | 36.90 | 37.04 | **41.49** | 61.26 | 46.74 | 68.30 | 63.98 | 57.06 | 56.76 | 57.92 | 38.34 | 52.08 | 50.94 | **73.12** |
| CongressEW | 60.64 | 0.37 | -1.66 | 0.27 | 23.39 | 25.46 | 58.34 | 53.63 | 60.64 | 61.35 | **69.54** | 53.11 | 0.31 | 2.15 | 0.64 | 28.31 | 30.98 | 50.71 | 43.29 | 53.11 | 52.65 | **59.80** | 88.97 | 61.61 | 58.62 | 61.61 | 54.48 | 58.16 | 87.82 | 86.67 | 88.97 | 89.20 | **91.72** |
| mfeat-zer | 62.70 | 52.42 | 18.02 | 25.68 | 41.19 | 58.83 | 61.21 | 36.82 | 56.05 | 52.08 | **66.33** | 71.52 | 61.91 | 48.87 | 51.93 | 61.54 | 69.09 | 71.14 | 54.51 | 64.92 | 68.37 | **72.41** | 76.75 | 68.40 | 36.50 | 50.50 | 59.60 | 68.85 | 71.65 | 40.40 | 70.50 | 64.25 | **78.65** |
| ionosphereEW | 3.96 | 1.82 | 12.43 | NA | 24.82 | 25.16 | 22.88 | 26.29 | 13.62 | 15.79 | **49.15** | 6.92 | 3.30 | 11.72 | NA | 24.90 | 19.11 | 20.06 | 18.83 | 9.73 | 11.03 | **41.39** | 72.36 | 70.94 | 61.82 | NA | 72.36 | 70.94 | 61.82 | 76.64 | 68.66 | 70.09 | **85.47** |
| banknote | 82.72 | 62.85 | 1.27 | -0.99 | 70.50 | 95.39 | 69.61 | 10.15 | 62.62 | 46.30 | **96.24** | 74.40 | 61.28 | 4.55 | 7.60 | 71.60 | 91.94 | 70.71 | 12.91 | 61.11 | 39.36 | **93.17** | 95.48 | 89.65 | 57.73 | 50.07 | 83.97 | 98.83 | 80.39 | 66.40 | 89.58 | 84.04 | **99.05** |
| isolet1234 | 53.49 | 21.66 | NA | 24.32 | 16.52 | 47.24 | 52.74 | 3.80 | 55.27 | 50.17 | **59.57** | 75.61 | 56.17 | NA | 60.75 | 57.69 | 70.63 | 74.24 | 22.80 | 76.03 | 76.27 | **77.91** | 59.12 | 30.23 | NA | 39.71 | 30.81 | 51.65 | 57.45 | 13.29 | 60.40 | 55.24 | **60.44** |
| Leukemia | 73.76 | -5.82 | -5.82 | -4.83 | 43.72 | 68.98 | 72.27 | 36.21 | 73.71 | 73.71 | **83.72** | 69.13 | 10.26 | 10.26 | 6.90 | 48.10 | 54.90 | 59.72 | 26.38 | 63.92 | 63.92 | **73.99** | 93.06 | 54.17 | 54.17 | 58.33 | 83.33 | 91.67 | 90.28 | 80.56 | 93.06 | 93.06 | **95.83** |
| MNIST(AE) | 72.83 | 45.37 | 30.75 | 26.33 | 57.15 | 65.15 | 71.08 | 23.01 | 73.80 | 59.66 | **83.40** | 80.61 | 56.00 | 59.43 | 50.50 | 71.19 | 74.24 | 76.60 | 41.20 | 77.70 | 77.35 | **83.95** | 78.29 | 59.77 | 45.39 | 36.05 | 65.43 | 72.21 | 77.03 | 39.43 | 85.68 | 64.91 | **92.01** |
| Umist(AE) | 68.46 | 50.50 | 48.91 | 46.89 | 37.45 | 62.58 | 71.66 | 39.61 | 69.79 | 50.84 | **82.88** | 88.49 | 77.13 | 78.04 | 78.04 | 71.89 | 84.14 | 86.14 | 68.47 | 89.16 | 77.67 | **91.87** | 73.04 | 55.13 | 56.00 | 60.35 | 45.57 | 63.13 | 74.96 | 47.65 | 74.43 | 58.96 | **85.22** |

*Figure 5.* Results on 16 real-world datasets (%, bold represents the best for a dataset).

tional spectral clustering based on $k$ nearest neighbor graph)

We evaluated these algorithms using standard clustering metrics: Adjusted Rand Index (ARI; (Steinley, 2004)), Normalized Mutual Information (NMI; (Xu et al., 2003)), and Accuracy (ACC; (Yang et al., 2010)).

For fair comparison, all algorithms were tuned to optimal hyperparameters. For TANGO, the neighborhood size $k$ was searched from 2 to 100 with a step size of 1. Algorithms requiring specification of the number of clusters used the ground-truth number of clusters. The detailed hyperparameter settings of the other algorithms are given in Appendix C.1.

Experiments were conducted on 14 UCI datasets and 2 image datasets (Table 1). All datasets were min-max normalized. For image datasets MNIST and Umist, we used AutoEncoder (AE) to reconstruct them into 64-dimensional representations. Fig. 5 presents the results of TANGO and 10 comparison algorithms on 16 real-world datasets. The detailed explanation has been shown in Appendix C.2.

We also obtained p $< 0.05$ from Friedman's test, and the pairwise mean rank differences between TANGO and other comparison algorithms surpassed the critical difference threshold (CD $= 2.30$) in subsequent Nemenyi's test, indicating the superior performance of TANGO is not due to random chance (see Appendix C.3).

To further validate the effectiveness and computational efficiency of TANGO, we also apply it to image segmentation using Berkeley Segmentation Dataset Benchmark and provide corresponding running times. We transform each pixel into a 5-dimensional vector, where two coordinates correspond to the spatial location of the pixel and three coordi-

nates represent the RGB color channels. The segmentation is done by clustering this 5-dimensional dataset (154 401 pixels). See Appendix C.4.

The effects of the hyperparameter $k$ have been analyzed in Appendix C.5, and the ablation study of TANGO has been shown in Appendix C.6. We also provide a discussion on the limitation about TANGO in Appendix C.7.2.

## 6. Conclusion and Future Work

In this paper, we first proposed a global-view measure, i.e., typicality, to quantify the confidence for a point to be a mode, to resolve the issue of current mode-seeking methods that they require case-by-case threshold settings or human intervention to identify modes. We also devised an effective and efficient algorithm to calculate the typicality of points, and provide theoretical analysis about it. Furthermore, the clustering method TANGO was designed by leveraging typicality to detect modes and form sub-clusters, and utilizing graph-cut with an improved path-based similarity. Extensive experiments on multiple synthetic and real-world datasets demonstrated the effectiveness and superiority of TANGO over state-of-the-art clustering algorithms.

Future research could explore methods to automatically determine the number of nearest neighbors $k$, and to investigate the effects of other types of density-ascending dependency and typicality.

## Acknowledgments

We would like to thank the anonymous reviewers for their invaluable help to improve the paper. This work was sup-

ported by the National Natural Science Foundation of China under Grant numbers 61806170 and 62276218, and the Fundamental Research Funds for the Central Universities under Grant numbers 2682022ZTPY082 and 2682023ZTPY027.

## Impact Statement

This paper presents work whose goal is to advance the field of Machine Learning. There are many potential societal consequences of our work, none which we feel must be specifically highlighted here.

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

# A. Proofs of Theorems

### A.1. Proof of Theorem 1

*Proof.* It suffices to prove that $I - B^{\mathsf{T}}$ is invertible for $B$ defined by Eq. 6. To demonstrate this, we show the existence of a full-rank square matrix $R$ such that $R(I - B^{\mathsf{T}})R^{\mathsf{T}}$ is also full-rank.

In fact, by sorting data points $x_1, \ldots, x_n$ in ascending order of density $\rho$, the rows and columns of dependency matrix $B$ will be reordered into a new matrix $M$. $M$ can be equivalently obtained by performing row transformations given by a full-rank square matrix $R$, and column transformations given by $R^{\mathsf{T}}$ (i.e., $M = RBR^{\mathsf{T}}$), where $R = r_1 \cdot r_2 \cdot \ldots \cdot r_t$, and each $r_i$ refers to an elementary row permutation matrix swapping two rows.

Note that each $r_i$ is an elementary row permutation matrix swapping two rows. It holds that $r_i \cdot r_i = I$ (swapping the same two rows of the identity matrix twice restores the identity matrix) and $r_i = r_i^{\mathsf{T}}$ (swapping two rows of the identity matrix yields the same matrix as swapping its corresponding two columns), thus $r_i \cdot r_i^{\mathsf{T}} = I$, and it is easy to verify $RR^{\mathsf{T}} = I$.

Therefore, we have $R(I - B^{\mathsf{T}})R^{\mathsf{T}} = RR^{\mathsf{T}} - RB^{\mathsf{T}}R^{\mathsf{T}} = I - M^{\mathsf{T}}$. Note that $M$ represents the reordered dependency matrix $B$ based on sorting data points in ascending order of density, thus $M_{ii} = 0$ since each $x_i$ cannot be its own leader, and for $j > i$, $M_{ji} = 0$ because $\rho(x_i) \leq \rho(x_j)$ implying $x_i$ cannot be the leader of $x_j$. Therefore, $M$ is an upper triangular matrix with zeros on the diagonal, and $R(I - B^{\mathsf{T}})R^{\mathsf{T}} = I - M^{\mathsf{T}}$ is a lower triangular matrix with ones on the diagonal, which is obviously full-rank.

Thus, $I - B^{\mathsf{T}}$ is full-rank and invertible, and the equation $T = B^{\mathsf{T}}T + \rho$ has a unique solution $T = (I - B^{\mathsf{T}})^{-1}\rho$. $\qquad\square$

### A.2. Proof of Theorem 3

*Proof.* We first construct an undirected weighted multigraph $M = (V, E, W)$ according to Definition 5, where:

- $V$ is the set of vertices, with each vertex $V_i$ representing a sub-cluster $G_i$;

- $E$ is the set of undirected edges, where an edge $e = (G_i, G_j, p, q) \in E$ if there is an edge in $G$ connecting data points $p$ and $q$ from two different sub-clusters $G_i$ and $G_j$ respectively;

- $W$ represents the weight of each edge in $E$. For any $e = (G_i, G_j, p, q) \in E$ connecting data points $p \in G_i$ and $q \in G_j$, we have $W(e) = C(p, q)$.

Note that $C(p, q) = 1$ for any $p, q$ from the same sub-cluster. Thus, the path-based similarity $\mathsf{PBSim}(r_i, r_j)$ between any two sub-clusters $G_i$ and $G_j$ is always determined by edges in $E$, and all data points from the same sub-cluster $G_t$ can be seen as a whole (vertex $V_t$ in $V$), and $\mathsf{PBSim}(r_i, r_j)$ can be calculated by conducting a variant of Kruskal's algorithm (Fischer et al., 2003) on $M$: gradually adding edges from $E$ to a new graph $M' = (V, \varnothing, \varnothing)$ in descending order of $W$, and setting $\mathsf{PBSim}(r_i, r_j)$ as $W(e)$ iff $e \in E$ the first added edge that makes $G_i$ and $G_j$ belong to the same connected component of $M'$.

The above time complexity is $O(|E| \log |E| + |V|^2)$ (Fischer et al., 2003). As $|E| \leq nk$ and $|V| \ll n$, the overall time complexity will be no more than $O(nk \log(nk))$. $\qquad\square$

# B. Tail behavior of Typicality

We can also model the typicality as a solution of distributional identity, and analyze the tail behavior of typicality from this perspective. Let us rewrite Eq. (1) as the following distributional identity:

$$T \overset{D}{=} \sum_{j=1}^{N} B_j T_j + \rho \tag{8}$$

where $N$ is the in-degree of points in dependency graph (number of points that consider a randomly chosen point in the dataset as leader), and it is an integer-valued random variable; $\overset{D}{=}$ denotes equality in distribution; $T_j$s are independent (since the dependency is hierarchical) and distributed as $T$; $B_j$s are independent and distributed as some random variable $B$; $T_j$s, $B_j$s, $N$ and $\rho$ are independent.

As the in-degree distribution in a graph nearly follows the power law behavior (Adamic & Huberman, 2000), we can model the in-degree in dependency graph as $N = N(X)$, where $X$ is regularly varying with index $\alpha_N > 1$ and $N(x)$ is the number of Poisson arrivals during the time interval $[0, x]$, when the arrival rate is 1. Thus, $N(X)$ is also regularly varying asymptotically identical to $X$ (Litvak et al., 2007):

$$P(N(X) > x) \sim x^{-\alpha_N} L_N(x) \text{ as } x \to \infty. \tag{9}$$

where $L_N(x)$ is a slowly varying function.

As Theorem 1 has already shown the existence and uniqueness of the nontrivial solution of Eq. (8), we then have the following conclusion derived from (Volkovich & Litvak, 2010):

**Theorem 4.** *If $B < 1$ and $\mathrm{P}(\rho > x) = o(\mathrm{P}(N > x))$ as $x \to \infty$, then*

$$\mathrm{P}(T > x) \sim C_N x^{-\alpha_N} L_N(x) \text{ as } x \to \infty \tag{10}$$

*where $C_N = (\mathrm{E}(B))^{\alpha_N}[1 - \mathrm{E}(N)\mathrm{E}(B^{\alpha_N})]^{-1}$*

Theorem 4 can be proven in a similar manner as in (Volkovich & Litvak, 2010), by using Laplace–Stieltjes transform and Tauberian theorem. Since $B$ and $\rho$ only range from 0 to 1, the assumptions in Theorem 4 are trivially satisfied in our scenario, indicating that the tail of typicality is strictly following the power law behavior with the same exponent as $N$.

## C. Detailed Experimental Results and Analyses

### C.1. Hyperpamameter Settings of the Algorithms

The neighborhood size $k$ was searched from 2 to 100 with a step size of 1 in TANGO, LDP-SC, DEMOS, QKSPP, DPC-DBFN, and KNN-SC. Noise ratio parameters in DCDP-ASC and NDP-Kmeans were searched from 0 to 0.2 with a step size of 0.001. The minimum proportion of points that each cluster must contain in LDP-MST was fixed at 0.018 as described in the article. DCDP-ASC and DEMOS required manual selection of core points on decision graphs, and we used the best-performing selection for each parameter setting. In QKSPP, density fluctuation parameter $\beta$ was searched from 0 to 1 with a step size of 0.1 for each neighborhood size $k$. CPF searched for $k$ and $\beta$ parameters among preset combinations provided in the source code. USPEC searched for the number of representative points with a step size of 100 from 100 to 1000 (or the nearest multiple of 100 below the maximum number of data points), and $k$ was searched for all possible values under each representative point count.

### C.2. Clustering on Real-world Datasets

Fig. 5 and Table 2 show the best experimental results and corresponding hyperparameter settings of TANGO and comparison algorithms on 16 real-world datasets. Specifically, LDP-MST on isolet1234 and NDP-Kmeans on ionosphereEW did not produce results due to the insufficient number of local density peaks compared to the target number of clusters. TANGO demonstrates superior performance across all datasets, by integrating local and global features of the data and employing a graph-cut scheme.

Especially on datasets such as banknote, Leukemia, and MNIST (AE), TANGO outperforms other algorithms by a significant margin in various metrics. Algorithms based on local density peaks such as LDP-SC, DCDP-ASC, LDP-MST, NDP-Kmeans, and DEMOS performed poorly on multiple datasets, particularly DCDP-ASC and LDP-MST, which often yielded clustering results close to random division. This demonstrates the challenge of achieving satisfactory clustering results using only local data features. USPEC demonstrates strong clustering ability, ranking second after TANGO on multiple datasets, but it notably underperforms on datasets such as semeion and ver2 compared to TANGO. Moreover, USPEC requires extensive parameter tuning, limiting its practical utility.

Mode-seeking algorithms like CPF and QKSPP also exhibit significantly weaker performance compared to TANGO, highlighting the effectiveness of TANGO for detecting modes through typicality and conducting graph-cut.

Fig. 6 illustrates breaking in density-ascending dependency identified by TANGO on the isolet1234 dataset (visualized in 2D using t-SNE for dimensionality reduction). The ability of TANGO to detect modes more effectively contributes to improved final clustering results.

| Dataset | Metric | LDP-SC | DCDP-ASC | LDP-MST | NDP-Kmeans | DEMOS | CPF | QKSPP | DPC-DBFN | USPEC | KNN-SC | TANGO |
|---|---|---|---|---|---|---|---|---|---|---|---|---|
| wdbc | ARI | 76.1 | 0.24 | 28.72 | 44.02 | 49.12 | 43.81 | 52.46 | 79.89 | 79.26 | 81.76 | **85.01** |
| | NMI | 65.4 | 0.52 | 31.91 | 43.88 | 47.82 | 39.38 | 42.94 | 69.14 | 68.33 | 73.89 | **77.35** |
| | ACC | 93.67 | 62.92 | 77.86 | 83.66 | 85.41 | 80.84 | 76.45 | 94.73 | 94.55 | 95.25 | **96.13** |
| | Par. | 4 | 0 | / | 0 | 2 | 12 / 0.1 | 13 / 0.7 | 198 | 13 / 300 | 9 | 86 |
| heartEW | ARI | 20.8 | 9.67 | -0.19 | 22.9 | 8.03 | 27.09 | 36.62 | 20.07 | 13.37 | 35.76 | **39.4** |
| | NMI | 16.26 | 14.13 | 2.04 | 17.72 | 12.72 | 21.24 | 28.87 | 21.57 | 9.54 | 27.95 | **30.7** |
| | ACC | 72.96 | 65.93 | 53.33 | 74.07 | 28.89 | 72.96 | 80.37 | 72.59 | 68.52 | 80 | **81.48** |
| | Par. | 9 | 0 | / | 0 | 16 | 12 / 0.1 | 30 / 0.4 | 13 | 4 / 100 | 42 | 35 |
| segmentation | ARI | 52.87 | 43.84 | 28.79 | 22.72 | 61.19 | 58 | 56.55 | 55.55 | 59.43 | 57.77 | **64.21** |
| | NMI | 66.16 | 61.6 | 59.13 | 55.38 | 70.4 | 71.83 | 68.17 | 69.99 | 70.34 | 68.6 | **71.87** |
| | ACC | 72.81 | 57.33 | 49.81 | 44.24 | 71.86 | 67.95 | 68.29 | 67.24 | 65.05 | 65.71 | **73.19** |
| | Par. | 10 | 0 | / | 0.2 | 46 | 64 / 0.3 | 50 / 0.9 | 5 | 28 / 1000 | 40 | 98 |
| semeion | ARI | 55.58 | 34.37 | 15.32 | 19.33 | 44.08 | 36.2 | 48.75 | 29.85 | 53.96 | 46.82 | **65.37** |
| | NMI | 66.15 | 49.53 | 45.57 | 40.36 | 63.53 | 52.69 | 64.85 | 41.37 | 66.71 | 61.24 | **71.89** |
| | ACC | 70.06 | 53.11 | 36.41 | 33.77 | 56.12 | 47.96 | 59.38 | 48.02 | 68.17 | 63.28 | **81.42** |
| | Par. | 18 | 0.2 | / | 0 | 20 | 14 / 0.1 | 13 / 0.0 | 3 | 6 / 1000 | 72 | 19 |
| semeionEW | ARI | 54.92 | 34.15 | 14.01 | 31.04 | 44.14 | 36.04 | 49.05 | 29.69 | 55.95 | 46.53 | **63.49** |
| | NMI | 67.33 | 47.36 | 43.49 | 46.21 | 63.54 | 52.14 | 64.17 | 42.04 | 67.64 | 60.49 | **71.11** |
| | ACC | 74.07 | 55.24 | 37.41 | 51.98 | 55.43 | 48.02 | 57.75 | 47.96 | 69.43 | 62.9 | **79.35** |
| | Par. | 4 | 0.2 | / | 0 | 20 | 14 / 0.1 | 13 / 0.0 | 3 | 3 / 1000 | 75 | 21 |
| ver2 | ARI | 22.92 | -0.66 | 33.35 | -3.1 | 28.38 | 34.82 | 37.62 | 33.17 | 31.54 | 25.27 | **42.85** |
| | NMI | 28.03 | 1.29 | 27.4 | 6.53 | 30.24 | 30.32 | 32.22 | 26.96 | 30.31 | 30.19 | **32.41** |
| | ACC | 55.16 | 47.42 | 64.52 | 41.94 | 64.84 | 52.58 | 69.03 | 69.68 | 60.32 | 50 | **73.55** |
| | Par. | 5 | 0 | / | 0 | 18 | 14 / 1.0 | 20 / 0.1 | 67 | 5 / 100 | 190 | 13 |
| synthetic-control | ARI | 61.58 | 60.03 | 40.1 | 51.24 | 58.18 | 66.33 | 62.73 | 55.53 | 68.86 | 65.02 | **72.23** |
| | NMI | 80.06 | 75.11 | 65.56 | 66.68 | 74.58 | 81.92 | 79.73 | 72.78 | 82.22 | 81.29 | **84.65** |
| | ACC | 57.33 | 64 | 38.17 | 55.5 | 64 | 66.67 | 62.5 | 59.5 | 71.67 | 68.17 | **80** |
| | Par. | 7 | 0 | / | 0.2 | 24 | 82 / 0.1 | 39 / 0.9 | 13 | 5 / 200 | 24 | 18 |
| waveform | ARI | 28.35 | 24.56 | 31.34 | 28.09 | 31.02 | 30.74 | 29.85 | 0.65 | 25.17 | 25.22 | **39.77** |
| | NMI | 36.31 | 35.64 | 34.88 | 29.98 | 39.25 | 39.14 | 36.84 | 5.63 | 36.9 | 37.04 | **41.49** |
| | ACC | 61.26 | 46.74 | 68.3 | 63.98 | 57.06 | 56.76 | 57.92 | 38.34 | 52.08 | 50.94 | **73.12** |
| | Par. | 27 | 0.2 | / | 0 | 71 | 65 / 0.8 | 38 / 0.5 | 2 | 11 / 1000 | 14 | 50 |
| CongressEW | ARI | 60.64 | 0.37 | -1.66 | 0.27 | 23.39 | 25.46 | 58.34 | 53.63 | 60.64 | 61.35 | **69.54** |
| | NMI | 53.11 | 0.31 | 2.15 | 0.64 | 28.31 | 30.98 | 50.71 | 43.29 | 53.11 | 52.65 | **59.8** |
| | ACC | 88.97 | 61.61 | 58.62 | 61.61 | 54.48 | 58.16 | 87.82 | 86.67 | 88.97 | 89.2 | **91.72** |
| | Par. | 8 | 0 | / | 0 | 26 | 21 / 0.1 | 41 / 0.7 | 2 | 6 / 300 | 16 | 9 |
| mfeat-zer | ARI | 62.7 | 52.42 | 18.02 | 25.68 | 41.19 | 58.83 | 61.21 | 36.82 | 56.05 | 52.08 | **66.33** |
| | NMI | 71.52 | 61.91 | 48.87 | 51.93 | 61.54 | 69.09 | 71.14 | 54.51 | 64.92 | 68.37 | **72.41** |
| | ACC | 76.75 | 68.4 | 36.5 | 50.5 | 59.6 | 68.85 | 71.65 | 40.4 | 70.5 | 64.25 | **78.65** |
| | Par. | 10 | 0 | / | 0 | 8 | 32 / 0.8 | 27 / 0.1 | 5 | 20 / 1000 | 4 | 40 |
| ionosphereEW | ARI | 3.96 | 1.82 | 12.43 | / | 24.82 | 25.16 | 22.88 | 26.29 | 13.62 | 15.79 | **49.15** |
| | NMI | 6.92 | 3.3 | 11.72 | / | 24.9 | 19.11 | 20.06 | 18.83 | 9.73 | 11.03 | **41.39** |
| | ACC | 60.68 | 65.24 | 67.81 | / | 72.36 | 70.94 | 61.82 | 76.64 | 68.66 | 70.09 | **85.47** |
| | Par. | 3 | 0 | / | / | 4 | 50 / 0.1 | 50 / 0.7 | 37 | 8 / 200 | 6 | 15 |
| banknote | ARI | 82.72 | 62.85 | 1.27 | -0.99 | 70.5 | 95.39 | 69.61 | 10.15 | 62.62 | 46.3 | **96.53** |
| | NMI | 74.4 | 61.28 | 4.55 | 7.6 | 71.6 | 91.94 | 70.71 | 12.91 | 61.11 | 39.36 | **93.59** |
| | ACC | 95.48 | 89.65 | 57.73 | 50.07 | 83.97 | 98.83 | 80.39 | 66.4 | 89.58 | 84.04 | **99.13** |
| | Par. | 13 | 0 | / | 0 | 4 | 65 / 0.3 | 29 / 0.8 | 83 | 10 / 1000 | 13 | 66 |
| isolet1234 | ARI | 53.49 | 21.66 | / | 24.32 | 16.52 | 47.24 | 52.74 | 3.8 | 55.27 | 50.17 | **59.57** |
| | NMI | 75.61 | 56.17 | / | 60.75 | 57.69 | 70.63 | 74.24 | 22.8 | 76.03 | 76.27 | **77.91** |
| | ACC | 59.12 | 30.23 | / | 39.71 | 30.81 | 51.65 | 57.45 | 13.29 | 60.4 | 55.24 | **60.44** |
| | Par. | 4 | 0 | / | 0 | 9 | 21 / 0.4 | 28 / 0.8 | 23 | 18 / 1000 | 26 | 48 |
| Leukemia | ARI | 73.76 | -5.82 | -5.82 | -4.83 | 43.72 | 68.98 | 72.27 | 36.21 | 73.71 | 73.71 | **83.72** |
| | NMI | 69.13 | 10.26 | 10.26 | 6.9 | 48.1 | 54.9 | 59.72 | 26.38 | 63.92 | 63.92 | **73.99** |
| | ACC | 93.06 | 54.17 | 54.17 | 58.33 | 83.33 | 91.67 | 90.28 | 80.56 | 93.06 | 93.06 | **95.83** |
| | Par. | 6 | 0 | / | 0 | 2 | 12 / 0.1 | 12 / 0.0 | 20 | 6 / 70 | 6 | 10 |
| MNIST(AE) | ARI | 72.83 | 45.37 | 30.75 | 26.33 | 57.15 | 65.15 | 71.08 | 23.01 | 73.8 | 59.66 | **83.4** |
| | NMI | 80.61 | 56 | 59.43 | 50.5 | 71.19 | 74.24 | 76.6 | 41.2 | 77.7 | 77.35 | **83.95** |
| | ACC | 78.29 | 59.77 | 45.39 | 36.05 | 65.43 | 72.21 | 77.03 | 39.43 | 85.68 | 64.91 | **92.01** |
| | Par. | 4 | 0.2 | / | 0 | 100 | 21 / 0.4 | 22 / 0.9 | 2 | 3 / 1000 | 9 | 33 |
| Umist(AE) | ARI | 68.46 | 50.5 | 48.91 | 46.89 | 37.45 | 62.58 | 71.66 | 39.61 | 69.79 | 50.84 | **82.88** |
| | NMI | 88.49 | 77.13 | 78.04 | 78.04 | 71.89 | 84.14 | 86.14 | 68.47 | 89.16 | 77.67 | **91.87** |
| | ACC | 73.04 | 55.13 | 56 | 60.35 | 45.57 | 63.13 | 74.96 | 47.65 | 74.43 | 58.96 | **85.22** |
| | Par. | 7 | 0.2 | / | 0.2 | 24 | 10 / 0.5 | 12 / 0.1 | 5 | 3 / 300 | 8 | 6 |

*Table 2.* Results on 16 real-world datasets (%, bold represents the best for a dataset, "Par." represents parameter settings).

## C.3. Statistical Significance Test Results

We first employ Friedman's test in experimental results in terms of ARI to determine if there exhibits significant diffrences among the performance of all algorithms. After confirming that, the Nemenyi's test is conducted as a post-hoc test to identify statistical significant diffrences between pairs of algorithms.

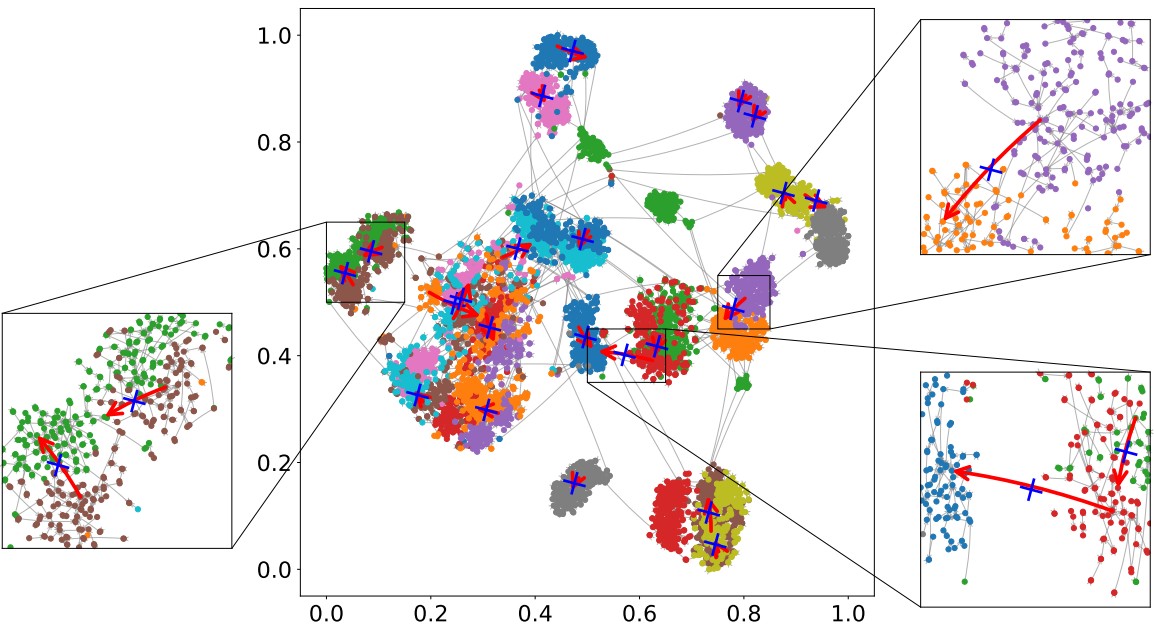

*Figure 6.* Illustration of TANGO breaking some dependencies based on typicality for the isolet1234 dataset (visualized in 2D by t-SNE).

We have obtained Friedman Statistic $= 96.4$ and $\text{p} = 2.9e - 16 < 0.05$ from Friedman's test, indicating significant differences in performance among the algorithms.

In subsequent Nemenyi's test, the critical difference threshold $\text{CD} = 2.30$ is calculated by $CD = q_\alpha \times \sqrt{\frac{K(K+1)}{6N}}$, where $K$ indicates the number of algorithms and $N$ the number of datasets, and $q_\alpha = q_{0.05} = 1.96$. The pairwise differences between algorithms are shown in Fig. 7 in the form of heatmap. Each cell in Fig. 7 represents the difference between corresponding pair of algorithms, and the larger (smaller) the value, the closer it is to red (blue). It can be observed that the differences between TANGO and all the other algorithms not only surpass CD, but also significantly exceed the extent of differences among the other algorithms.

### C.4. Image Segmentation and Running Times

Fig 8 shows the segmentation results of different clustering methods on 6 images from Berkeley Segmentation Dataset Benchmark, as well as corresponding running times (see the number above each image), where each image is a dataset containing $154\,401$ samples (each sample refers to a pixel). We use image segmentation as an extension to test the efficiency of TANGO on larger datasets, and also as a preliminary result (without fine-tuning the parameters) to demonstrate its promising application to other tasks. The performance of TANGO varied for different images, with overall better performance on several images (row 1, row 2, and row 5) but some flaws in some areas of other images. Note that in row 3, although the cloth and the hand blend for TANGO, it is the only method that successfully segments features in the face (mouth and eyes).

For TANGO, we also show the running time of the similarity matrix calculation (parallelized with 20 threads) in parentheses, which is the main cost of the overall algorithm and can be easily parallelized. It has also been shown that the remaining part of TANGO is highly efficient (taking no more than 4 seconds across each image), which aligns with the theorems about efficiency.

The experimental environment is: Windows 11, Python 3.11, CPU i7-13700KF and 32GB RAM.

### C.5. Effects of the Hyperparameter k

The proposed TANGO algorithm has only one hyperparameter, which is the number of nearest neighbors $k$. Fig. 9 illustrates the performance of TANGO and three other density-based algorithms on six real-world datasets as the local

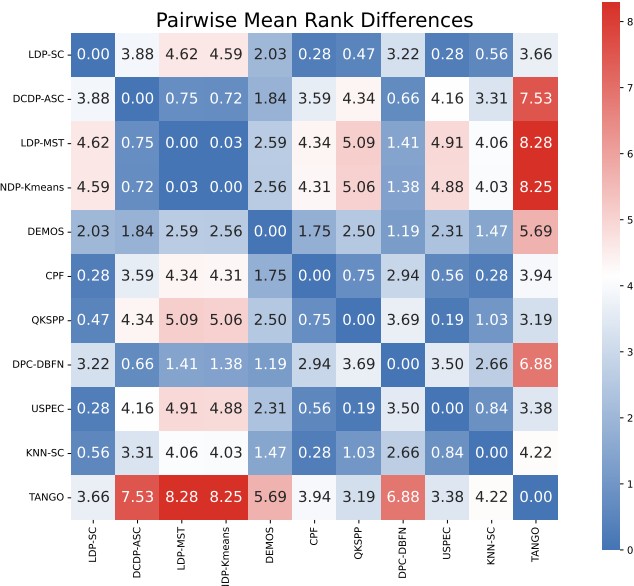

*Figure 7.* Statistical significance test.

neighborhood parameter $k$ varies. Here, the additional density fluctuation parameters of QKSPP and CPF are fixed at $0.7$ and $0.5$, respectively. From the figure, it is observed that the performance of TANGO tends to stabilize as $k$ increases and demonstrates strong overall effectiveness, whereas DPC-DBFN, QKSPP, and CPF exhibit significant fluctuations within the same parameter range. Moreover, in practical applications, optimizing the density fluctuation parameters of QKSPP and CPF requires fine-tuning to achieve optimal clustering results, thus TANGO demonstrates advantages in hyperparameter stability and practicability.

For TANGO, the parameter $k$ affects the similarity measure. When $k$ is small, the similarity may not be comprehensive enough to capture the complex distribution around two points, thus increasing $k$ can lead to better performance. When $k$ is relatively large, increasing $k$ will introduce new shared nearest neighbors of two data points $x_i$ and $x_j$, which, however, will have relatively small contribution to the similarity as these neighbors have large distance to both $x_i$ and $x_j$, and similarity values become stable. In this case, the subsequent process of the algorithm will have similar results and thus the performance will also become stable. Based on this, we recommend a larger $k$ for datasets with complex distribution or having a large number of points, to more comprehensively capture the neighboring information around two points.

### C.6. Ablation Study

To validate the effectiveness of the main components of TANGO, we conducted ablation experiments by testing the performance after removing certain components:

- TANGO-a: Removing the whole typicality-aware mode-seeking step and just directly applying spectral clustering based on path-based similarity (Eq. (7)) to all data points. This is to show that the typicality-aware mode-seeking step is essential.

- TANGO-b: Including the mode-seeking step but not typicality-aware, to further validate the significance of typicality in mode-seeking procedure.

- TANGO-c: Removing the aggregation step on mode-centered sub-clusters. This is to validate that further aggregating the sub-clusters into the final partition with the specified number of clusters is necessary.

Fig. 10 illustrates the results of the ablation experiments. TANGO-a suggests that identifying representative sub-clusters by mode-seeking first can provide a more robust foundation than directly clustering all data points. TANGO-b performed better than TANGO-a but worse than the full TANGO, indicating that while the general mode-seeking process is valuable,

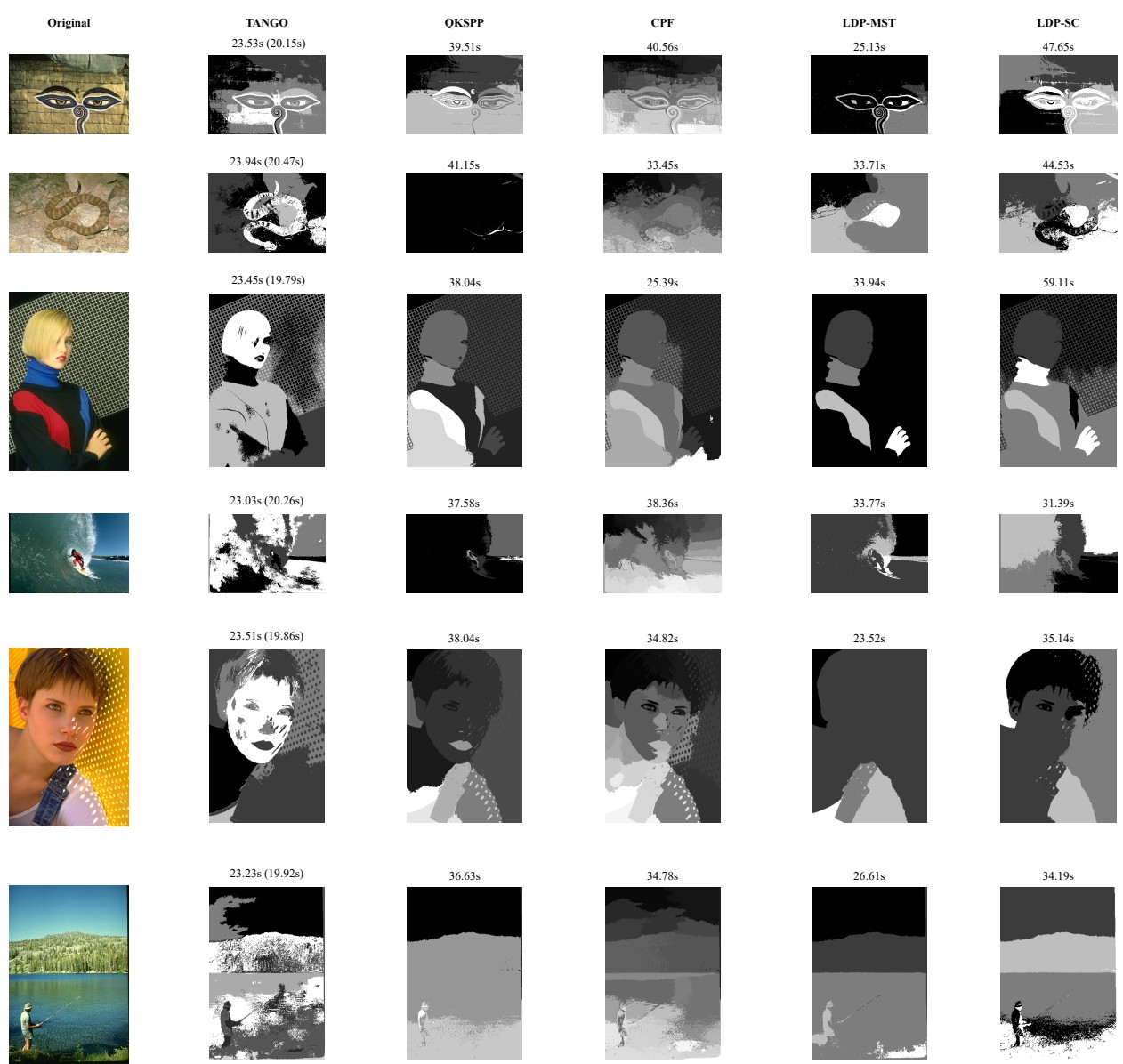

*Figure 8.* Results on Image Segmentation. We fixed $k = 50$ in TANGO and LDP-SC; $k = 300$ and $\beta = 0.9$ in Quick Shift++ and CPF; target number of clusters was fixed at 5 in TANGO, LDP-MST and LDP-SC.

incorporating typicality can further refine the sub-clusters and contribute to the final performance. From TANGO-c, we can see that it is essential to further aggregate the sub-clusters into final partition, especially when a specific number of clusters is required. Without aggregation, the mode-centered sub-clusters, though potentially meaningful locally, still fail to form the globally superior partitions.

Overall, these results highlight that while typicality-aware mode-seeking establishes strong initial sub-clusters, the aggregation step is also crucial to achieve a high-quality final result. TANGO's effectiveness stems from this multi-stage process.

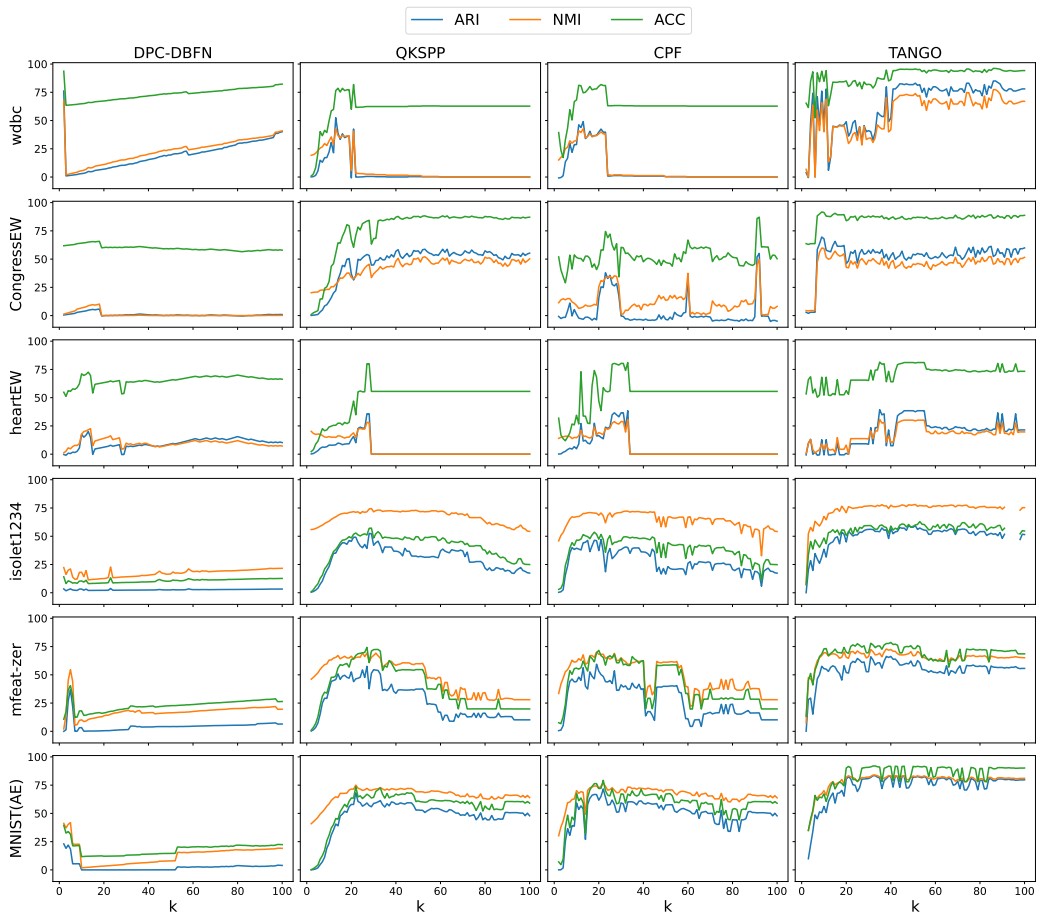

*Figure 9.* The impact of the number of neighbors $k$ on the performance of TANGO and other density-based algorithms.

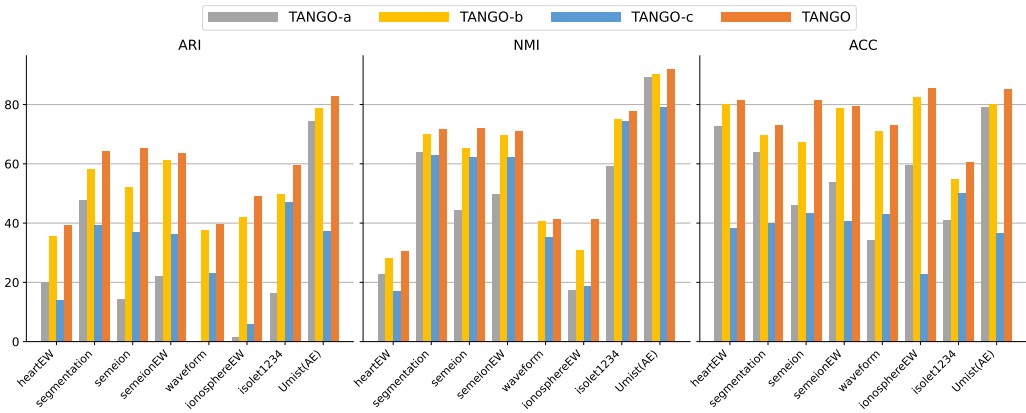

*Figure 10.* Results of ablation study.

## C.7. Additional Discussion

### C.7.1. VALIDATION ON HIGHLY NOISY DATASETS

Fig. 11 visualizes the clustering results for TANGO on four noisy datasets with irregular shapes. The results demonstrate that TANGO exhibits high performance and is robust to noise. This may be due to the mode-seeking step that can naturally

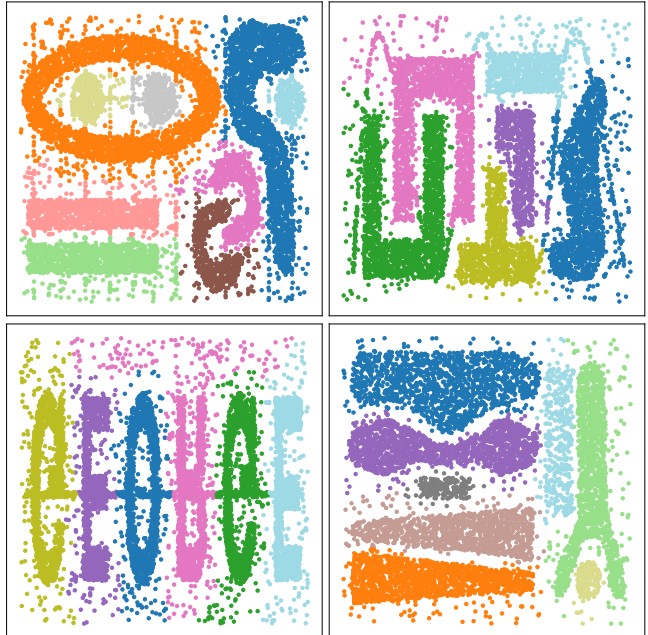

*Figure 11.* Clustering results for TANGO on four noisy synthetic datasets with irregular shapes.

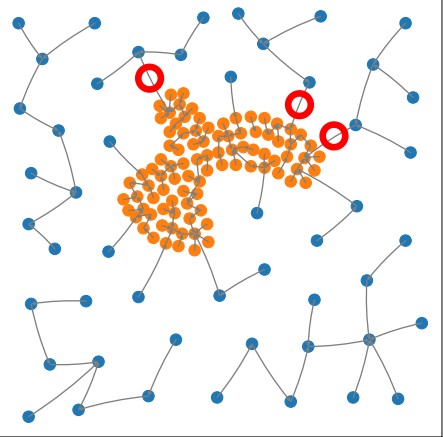

*Figure 12.* The final dependency connections after the typicality-aware mode-seeking process on a dataset. Some dependency was still wrongly maintained and would cause a significant decrease in performance. We used red circle to mark such dependency.

alleviate the impact of noise, and path-based spectral clustering has the advantage to capture complex data distribution.

### C.7.2. LIMITATION

One example of limitation is that TANGO would perform worse when dealing with the dataset in Fig. 12, where the low density points lie uniformly around the high density points with extreme density discrepancy between them. In such a case, the typicality collected by low density points, would be less than that of higher density ones, making the dependency connections between them not broken (marked by red circles). Future work could explore whether other types of dependency can address this situation.

