# OpenReview forum: "TANGO: Clustering with Typicality-Aware Nonlocal Mode-Seeking and Graph-Cut Optimization"
_ICML.cc/2025/Conference — ICML 2025 poster_

### Official Review · Reviewer_4KVW · 2025-03-08

**Overall Recommendation:** 4

**Summary:**

The paper introduces TANGO  (Typicality-Aware Nonlocal Mode-Seeking and Graph-Cut Optimization), a clustering algorithm that leverages typicality, a global measure of a point's confidence to be a mode, to address the limitations of traditional mode-seeking methods that rely on local data characteristics and case-by-case threshold settings. TANGO integrates typicality-aware mode-seeking with graph-cut optimization and an improved path-based similarity to aggregate data into clusters. Experimental results on synthetic and real-world datasets demonstrate TANGO's effectiveness and superiority over state-of-the-art clustering algorithms.


## update after rebuttal

Our evaluation remains unchanged after reviewing the rebuttal; the paper deserves an Accept (4) score.

**Claims And Evidence:**

Yes, the main claims in the paper are supported by clear and convincing evidence.

**Essential References Not Discussed:**

no

**Experimental Designs Or Analyses:**

Yes

**Methods And Evaluation Criteria:**

Yes, the proposed methods and evaluation criteria are well-suited for the problem and application at hand.

**Other Comments Or Suggestions:**

1. It is necessary to further explain how Typicality adjusts contributions through density-weighted mechanisms (e.g., rank-based dependencies in Eq. (6)), distinguishing it from PageRank’s uniform jump probability assumption. Additionally, clarify how the recursive formula (Eq. (1)) reflects the "attraction" mechanism of density peaks.
2. Add runtime comparisons between TANGO and other algorithms on datasets of varying scales in Table 2 or the appendix to validate the practical efficiency of the claimed time complexity $O(nk^2d)$.
3. Expand Figure 10 to include experiments isolating the contributions of sub-cluster generation (Typicality module), path-based similarity (PBSim), and spectral clustering, demonstrating the necessity of their synergy.
4. Analyze the fluctuations in TANGO’s performance with varying $k$ values in Figure 9
5. Provide 1–2 cases where TANGO underperforms to discuss its limitations.
6. Define the variable $p$ in $O(p^3 + n)$ (whether it aligns with the $p$ in Algorithm 3).
7. Discuss classical spectral clustering methods in the related work section.  Discuss why spectral clustering was chosen for merging sub-clusters, unlike hierarchical clustering in other DPC-based techniques.
8. The adoption of SNN-based density estimation indeed enhances the TANGO algorithm's performance, as it better captures local structures, delivering superior capability in generating sub-clusters. While other comparative algorithms likely rely on simpler density estimation methods, it remains to be verified whether the proposed algorithm would maintain its performance advantage, as demonstrated in the experiments, if the comparative algorithms (e.g., LDP-MST, DPC, DEMOS, LDP-SC) were to adopt equivalent density estimation techniques.
9. Replace the equality symbol in Line 253, “$T(x_j) = T(x_j) + T(x_i) \cdot B_{ij}$”, with the assignment operator “$T(x_j) \leftarrow T(x_j) + T(x_i) \cdot B_{ij}$”.
10. Add a footnote to the “Par.” column in Table 2 to clarify it represents “hyperparameter settings”.
11. Fix inconsistent citations (e.g., “Eq.2” → “Eq. (2)”).

**Other Strengths And Weaknesses:**

The paper’s strengths lie in its originality, introducing typicality as a global measure inspired by PageRank, and its significance, demonstrated through superior performance on diverse datasets and theoretical rigor. It is well-structured and clear, with effective illustrations. Weaknesses include sensitivity to the hyperparameter $k$, potential scalability issues with path-based similarity, and limited validation on highly noisy or imbalanced datasets.

**Questions For Authors:**

no

**Relation To Broader Scientific Literature:**

The paper advances clustering by introducing ​typicality, a global measure inspired by ​PageRank, to address limitations of local mode-seeking methods like ​Mean Shift and ​DPC. It integrates ​graph-cut optimization and ​path-based similarity, building on spectral clustering techniques. Theoretical analysis ensures efficiency, while experiments on diverse datasets demonstrate superiority over state-of-the-art methods, bridging local and global perspectives in clustering.

**Theoretical Claims:**

Yes

---

> ### Author Rebuttal · Authors · 2025-03-30
>
> Thank you so much for reviewing our paper. We answer your main concerns below.
>
> Concern about the validation on highly noisy or imbalanced datasets: TANGO can indeed perform well on noisy and imbalanced datasets such as "cluto-t4-8k", "cluto-t5-8k", "cluto-t8-8k", "cluto-t7-10k" and "unbalance". This may be due to the mode-seeking step that can naturally alleviate the impact of noise, and path-based spectral clustering has the advantage to capture complex data distribution. We can include these in the experiments as an additional discussion.
>
> Other Comments:
>
> 1: From the PageRank perspective, for each point $x_i$ and its nearest higher density neighbor (leader) $x_j$, $B_{ij}$ denotes the probability from $x_i$ jumping to $x_j$, and $x_i$ has the probability $(1-B_{ij})$ to stay. The right side of Figure 2 has shown how Eq. (1) reflects a density peak $x_i$ collecting typicality from all points in its "attraction". For example, as shown in the right side of Figure 2, when $T(x_i) = B_{1i} T(x_1) + B_{2i} T(x_2) + B_{3i} T(x_3) + \rho_i$ from Eq. (1), and $T(x_1)=\rho_1$, $T(x_2)=B_{42}T(x_4)+\rho_2$, $T(x_3)=\rho_3$, $T(x_4)=\rho_4$, also from Eq. (1) respectively, then $T(x_i)=B_{1i} \rho_1 + B_{2i} (B_{42}\rho_4+\rho_2) + B_{3i} \rho_3 + \rho_i$, indicating that $x_i$ collects typicality from $x_1$, $x_2$, $x_3$ and $x_4$, which are all points in its "attraction". We will make this clearer in the revision.
>
> 2: The datasets in Table 2 is not much larger, and most of the algorithms as well as TANGO can complete their execution within a very short time (no more than 3 seconds). That’s why we have further presented the experimental results and corresponding running times on image segmentation task, which is done by clustering a dataset containing 154,401 samples (each image is a dataset and each pixel is a sample, as mentioned in the right side of Line 411), to show the efficiency on larger datasets.
>
> 3: We will include these in ablation study.
>
> 4: We will include an analysis in the revision. In Figure 9, as $k$ increases, the clustering performance of TANGO initially rises and then stabilizes. The parameter $k$ affects the similarity measure. When $k$ is small, the similarity may not be comprehensive enough to capture the complex distribution around two points, thus increasing $k$ can lead to better performance. When $k$ is relatively large, increasing $k$ will introduce new shared nearest neighbors of two data points $x_i$ and $x_j$, which, however, will have relatively small contribution to the similarity as these neighbors have large distance to both $x_i$ and $x_j$, and similarity values become stable. In this case, the subsequent process of the algorithm will have similar results and thus the performance will also become stable.
>
> 5: One possible example of limitation is that TANGO would perform worse when dealing with the right side of the "Compound" dataset, where the low density points lie uniformly around the high density points with extreme density discrepancy between them. In such a case, the typicality of low density points would be less than that of higher density ones, making them become a single subcluster. Future work could explore whether other types of dependency can address this situation. We will include this discussion in the revision.
>
> 6: $p$ is the number of modes (subclusters) detected by Algorithm 2. We will make this more formal and clearer.
>
> 7: We will include more related works about spectral clustering in the revision. The reason why choosing spectral clustering is that it comprehensively considers a global graph-cut cost of the whole partition. On the other hand, hierarchical clustering partitions the data greedily and ignores the global impact on the whole partition at each greedy step, making it always achieve an inferior and imbalanced partition.
>
> 8: Thank you for the suggestion. We tested applying SNN-based similarity and corresponding density to LDP-MST, LDP-SC and DEMOS, but observed still less performance than TANGO, even worse than their original implementations in some cases. This might be due to that these methods didn't employ typicality to construct tree-like subclusters, and their aggregation approaches for the subclusters also missed some important information that can be revealed by path-based spectral clustering technique. By the way, SNN-based approach has also been used in LDP-MST and LDP-SC by their authors, to determine the similarity between subclusters.
>
> 9, 10 and 11: We will correct all the issues you have mentioned.

---

### Official Review · Reviewer_FABQ · 2025-03-08

**Overall Recommendation:** 3

**Summary:**

This paper introduced the notion of "typicality" in density-based clustering, which measures the likelihood or confidence that a certain point should be a mode (a center) of a cluster. Existing techniques determine modes based on local measures (e.g., density of a point), but the premise of the paper is that in some situations, global features of the dataset will dictate whether a point should really be a mode. The measure of typicality introduced is defined recursively depending on a points density and the typicality of other nearby points. With this measure in hand, the paper presents TANGO, a framework for density based clustering that incorporated the typicality measure with a path-based similarity measure and spectral clustering. Experiments are run on several datasets.

## Update after rebuttal

Thanks to the authors for the updates. I appreciate the larger experiments and the accompanying runtimes. Overall I still have a positive overall view of the paper.

**Claims And Evidence:**

The motivation for typically is well reasoned and Figure 1 is nice.

The experimental results do provide an indication that the method is outperforming other methods.


This is not major but there is a claim in the supplement that seems overstated. "As the in-degree distribution in a graph always follows the power law behavior..." This is overstated. There is evidence for power law distributions being strong but claiming that in-degree distributions "always" follow this distribution is not true.

Post rebuttal: thanks to the authors for acknowledging this concern.

**Essential References Not Discussed:**

None that I know of.

**Experimental Designs Or Analyses:**

I checked the experimental results and comparisons with other methods. The baselines, dataset, and cluster quality metric (e.g., ARI, MNI< ACC) are reasonable. For the results in Figure 5, it seems a little strange to optimally tune hyperparameters. While this might help in ways with making comparisons fair (by in some sense finding the best case scenario for each approach), it feels somewhat unnatural in that in practical applications one typically cannot tune all hyperparameters in this way. So this somewhat hides one of the difficult parts of running these methods. Nevertheless, this not a major concern since at least this was done for all methods.

**Methods And Evaluation Criteria:**

Strengths:

* The definition of typicality and the overall methodological approach of TANGO seems reasonable
* As far as evaluation, I appreciate the additional detail in the supplement on ablation study, the study on the effects of the hyperparamter k, and the image segmentation experiments.
* The evaluation across several different datasets and multiple baselines in Figure 5 is good

Weaknesses:

* The datasets considered are on the smaller side
* The paper mentions using spectral clustering, but there's more than one method that has been called spectral clustering and there can be many different variations of this. Would be better to state in more detail what is meant (e.g., computing top how many eigenvectors, and then clustering them with what? k-means? Or do you find low-conductance cut in the graph and use recursive bipartitioning?)

Post-rebuttal: thanks to the authors for clarifying what approach they use.

**Other Comments Or Suggestions:**

Just some minor suggestions:

* Typo: "garph" instead of graph in at least one place
* Figure 3 is good but far removed from Figure 1, which makes it cumbersome to page back and forth to get the point of this figure. Both Figures also have a lot of whitespace. You may be able to make Figure 3 smaller and just have it be a subfigure, and then use another subfigure to show the important part of Figure 1 again. Just a suggestion.
* In the preliminaries, the paper mentions similarities, the density of a point, and the dependency. The meaning of these becomes clear later but is not specified at first, making this a confusing read for me. It might be good to either define these more carefully up front, or at least mention to the reader that specific measures of similarity, density, and dependency would follow later to keep them from thinking they are missing something important about the setup.

**Other Strengths And Weaknesses:**

Strengths:

* The definition of typically is reasonable and the paper does a good job motivating it. I think Figure 1 is helpful and it's placement at the front of the paper is good.
* The fact that typically can be computed efficiently is a strength
* The paper overall is fairly well written and organized

Weaknesses:

* The paper mentions "some theoretical analysis" as a contribution but it's vague early on. When we get to the details, the theoretical analysis seems mostly to be straightforward results about the complexity of computing typicality and runtime results. It would be better if the authors simply stated up front what "theoretical analysis" they provide, if it is really a big contribution. Otherwise, it feels a bit like the paper is trying to get credit for establishing theoretical results without stating what theory is provided.
* TANGO seems to perform well in practice but the experimental results do not seem that extensive. The datasets considered are a bit on the small side.
* The fact that all choices of k were tried makes me worry about runtime.

**Questions For Authors:**

What are runtimes for the methods?

How expensive is it to run TANGO for so many choices of k?

How does TANGO perform on larger datasets?

**Relation To Broader Scientific Literature:**

Yes, the paper outlines the related work.

**Theoretical Claims:**

I did not check the proofs of the theoretical claims, but the theoretical claims are appear to be straightforward algorithm runtime results. There are no surprises with these results or concerns about correctness.

---

> ### Author Rebuttal · Authors · 2025-03-30
>
> Thank you so much for reviewing our paper. We answer your main concerns below.
>
> Overstatement about power law distribution: Thank you for pointing this out. We will correct it.
>
> Weakness 1: Thank you for the comment. To demonstrate scalability, we have expanded our evaluation to a substantially larger image segmentation dataset (each image is a dataset containing 154,401 samples, as mentioned in the right side of Line 411) in Appendix C.4, which has shown the promising results of TANGO on larger datasets and its efficiency. We will add more detailed analysis on the image segmentation experiments in the revision.
>
> Weakness 2: Thank you for pointing this out. Specifically, we use the spectral clustering method called Normalized Cut, which computes the Symmetric Normalized Laplacian of similarity matrix, and applies k-means++ on the rows of the matrix whose columns consist of the eigenvectors of Laplacian corresponding to the smallest $nc$ eigenvalues ($nc$ is the number of target clusters). For implementation, we used the "SpectralClustering" module from "scikit-learn".
>
> Other Comments and Weaknesses: Thank you for suggestions! We will address them in the revision.
>
> Question 1: For finding the optimal choice of $k$ in TANGO, we use "gp_minimize" from "scikit-optimize" to find the value $k$ that maximizes the ARI by iteratively selecting $k$ based on a Gaussian Process model and an acquisition function. In practice, we found that we can run just around 20 different values of $k$ from range 2 to 100, to find the optimal one.
>
> For Question 2, please refer to the response to Weakness 1.

---

> > ### Comment · Reviewer_FABQ · 2025-04-01
> >
> > Thanks for your reply and the additional details! This answers my question about your spectral clustering approach and clarifies how many values of k you use. Nice to hear you have some experiments on larger datasets.
> >
> > One thing that has still not been clarified for me is the running time in practice for this and competing methods.

---

> > > ### Author Response · Authors · 2025-04-02
> > >
> > > Thanks for your reply! We answer your concern about the practical running times in the following.
> > >
> > > For datasets in Table 2, most of the comparisons as well as TANGO can complete their execution within a very short time. That's why we further evaluated running times on the image segmention task, where each image is a dataset containing 154,401 samples and each sample refers to a pixel. Figure 8 in appendix also shows the running times for TANGO and 4 representative competing methods (see the numbers above each image). For TANGO, we also presented the running times of the similarity matrix calculation (parallelized with 20 threads) in parentheses, which is the main cost of the overall algorithm and can be easily parallelized, as described in Line 820. It can also be seen that the remaining part of TANGO is highly efficient, which aligns with the theorems about efficiency. We also present the running times in Figure 8 below, as well as some of the largest datasets in Table 2.
> > >
> > >
> > > |      | TANGO         | QKSPP  | CPF    | LDP-MST | LDP-SC |
> > > |:--------:|:---------------|--------:|:--------:|:---------:|:--------:|
> > > | Image 1 | 23.53s (20.15s) | 39.51s | 40.56s | 25.13s  | 47.65s |
> > > | Image 2 | 23.94s (20.47s) | 41.15s | 33.45s | 33.71s  | 44.53s |
> > > | Image 3 | 23.45s (19.79s) | 38.04s | 25.39s | 33.94s  | 59.11s |
> > > | Image 4 | 23.03s (20.26s) | 37.58s | 38.36s | 33.77s  | 31.39s |
> > > | Image 5 | 23.51s (19.86s) | 38.04s | 34.82s | 23.52s  | 35.14s |
> > > | Image 6 | 23.23s (19.92s) | 36.63s | 34.78s | 26.61s  | 34.19s |
> > > | MNIST(AE) | 7.31s (7.06s) | 3.51s | 12.43s | 2.47s  | 3.15s |
> > > | isolet1234 | 7.53s (7.24s) | 16.18s | 51.12s | / | 2.43s |
> > > | waveform | 6.77s (6.61s) | 0.42s | 1.35s | 0.42s  | 1.95s |

---

### Official Review · Reviewer_vHVP · 2025-03-14

**Overall Recommendation:** 2

**Summary:**

The paper introduces TANGO, a novel clustering algorithm that integrates typicality with graph-cut optimization. The primary contribution is the concept of typicality, a novel measure to quantify the confidence of a point being a mode for a cluster. Experimental results demonstrate the efficacy of the proposed algorithm.

**Claims And Evidence:**

Yes

**Essential References Not Discussed:**

No

**Experimental Designs Or Analyses:**

Yes, I have checked the whole experiment section.

**Methods And Evaluation Criteria:**

Yes

**Other Comments Or Suggestions:**

1.	Mention the method used to generate visualizations for the datasets (ex. UMAP/t-SNE).
2.	Line193, right column. The following statement is unclear and would help from rephrasing/breaking up: Therefore, we define a similarity measure based on shared nearest neighbors, where we distinguish the varying contribution to similarity of each shared neighbor to have better robustness.
3. Clarify the difference between TANGO (typicality) and TANGO (final) in Fig.4

**Other Strengths And Weaknesses:**

Strength:
1. The method is theoretically sound and well-motivated.
2. The remarks in between the definitions and other theories are refreshing and contribute to the explanations of these definitions/theorems.
3. The visualizations showing which exact dependencies are broken by typicality are helpful in seeing that TANGO does actually address the drawbacks of other methods such as Quick Shift/DPC.
4. The results presented are significant.

Weakness
1. It is unclear what the parameter k denotes in Line 331, Algorithm 3. k is used as nearest neighbours (Line 198), most similar density points (Line 211) and a number of hops parameter (Line 245).
2. The method takes the total/desired number of modes as a parameter (p), so it cannot find modes on its own.
3. It is explained why typicality as a measure is important but not why the proposed implementation through hierarchical dependencies is a good choice for typicality.
4. What does the dependency matrix B mean? The note on Line 265 should be expanded.
5. The path-based similarity between two sub-clusters G_i, G_j (Line 285) can be explained more clearly by formally defining C and explaining it to be the connectivity matrix before Definition 5.
6. Also, a more convincing reason (than being intuitive) for the connectivity formulation being what it is (max over all paths of min C in that path) would be appreciated.
7. Are the presented results reproducible?
8. The results are limited to only small datasets, with the largest being MNIST (10k nodes). It would be important to know if the results scale to very large datasets such as ogbn-arxiv/etc. as well.
9. TANGO performs qualitatively worse in the image segmentation results (Pg15). For example, in row 3, the cloth and the hand completely blend, and there are many “blemishes” in the segmentation mask compared to all the other methods. In row 4, TANGO completely fails to segment the wave.
10. The ablation study, while present, is not extensively discussed or analyzed.
11. In the ablation study, typicality appears to contribute only marginally to performance, with a low but noticeable uplift. How do its theoretical benefits translate into real-world advantages?

**Questions For Authors:**

Refer to weakness

**Relation To Broader Scientific Literature:**

TANGO builds on density-based clustering by introducing typicality, a global confidence measure to identify cluster modes without manual tuning, unlike DPC and Quick Shift. It integrates graph-cut optimization with a path-based similarity metric, improving upon spectral clustering and density-peak methods. TANGO achieves superior clustering performance, outperforming 10 state-of-the-art methods across 16 real-world datasets.

**Theoretical Claims:**

Yes, I have checked the proof of Theorem 1 and Theorem 2. The proof of Theorem 1 is confusing. The introduction of the
R matrix is not clear, and it is unclear why R is a symmetric matrix. The definitions are not clearly explained.

---

> ### Author Rebuttal · Authors · 2025-03-30
>
> Thank you so much for reviewing our paper. We answer your main concerns below.
>
> Weaknesses:
>
> 1: Sorry for the confusion. $k$ in Line 331 is the number of nearest neighbors to define similarity (Line 198) and density (Line 211), which are the same $k$ that is the input parameter of TANGO. $k$ in Line 245 is just related to the summation of infinite series $T = \sum_{k=0}^{\infty}(B^\intercal)^k\rho$, and has nothing to do with the former ones. We will make these notations clearer.
>
> 2: $p$ is not an input parameter but the number of modes (subclusters) automatically identified by Algorithm 2, and it is used to help the analysis of time complexity. We will make this more formal and clearer.
>
> 3: In the hierarchical dependency, each data point only links to its nearest higher density neighbor, and this is an effective and efficient "density hill-climbing" procedure to assign data points to their corresponding modes. It has been widely used in many density-based clustering methods such as Quick Shift, DPC, Quick Shift++, DEMOS, CPF, and LDP-MST. Its consistency guarantee has also been theoretically proven by several articles such as "On the consistency of quick shift" and "A theoretical analysis of density peaks clustering and the component-wise peak-finding algorithm". Future research could also explore other types of dependency, as described in the right side of Line 435.
>
> 4: In the dependency matrix $B$, each element $B_{ij}$ denotes the weight of dependency from point $x_i$ to $x_j$. The notation of $B$ is first specified in the Preliminaries section (Line 119, 151 and 152). We will also make the note on Line 265 more thorough.
>
> 5 and 6: Thank you for your suggestions. We will make the definition of $C$ more formal. The detailed clarification on the connectivity formulation is included in the proof of Theorem 3 in the appendix. We will make it more thorough in the main body of the revision.
>
> 7: We have provided the code and datasets in supplementary material for reviewer to validate its reproducibility. The parameter settings are also included in Table 2 in the appendix.
>
> 8: Thank you for your suggestion to evaluate our algorithm on larger datasets. In fact, we have already provided experiments on image segmentation task with 154,401 samples per image (each image is a dataset and each pixel is a sample, as mentioned in the right side of Line 411).
>
> 9: The performance of TANGO is indeed not perfect in the image segmentation task. However, we use this task as an extension to test the efficiency on larger datasets, and also as a preliminary result (without tuning the hyperparameters) to demonstrate its promising application for other tasks. The performance of TANGO varied for different images, with overall better performance on several images (row 1, row 2, and row 5) but some flaws in some areas of other images. Note that in row 3, although the cloth and the hand blend for TANGO, it is the only method that successfully segments features in the face (mouth and eyes).
>
> 11: In fact, the contribution of typicality can be observed by comparing TANGO-b and TANGO in the ablation study. There are significant performance decrements when the typicality component was removed, such as "semeion" (ARI from 65.37 to 52.18), "ionosphereEW" (from 49.15 to 39.13), "isolet1234" (from 59.57 to 49.75) and "Umist (AE)" (from 85.22 to 78.67). Figure 6 has also shown the real-world advantages of typicality.
>
> Other Comments:
>
> 1: We have already mentioned the t-SNE in Line 712 for Figure 6 and will make it clearer.
>
> 3: TANGO (typicality) is to visualize the breaking of dependency via typicality, and TANGO (final) represents the final clustering results by TANGO, as mentioned in the right side of Line 330. We will make this clearer in the revision.

---

### Official Review · Reviewer_viff · 2025-03-15

**Overall Recommendation:** 3

**Summary:**

The authors first propose a global perspective metric, typicality, to quantify the confidence of a point being a mode. This addresses the limitation of current mode-seeking methods, which require manually setting thresholds or human intervention to identify modes. They also design an efficient and effective algorithm to compute typicality and provide theoretical analysis. Furthermore, they introduce the TANGO clustering method, which leverages typicality to detect modes and form subclusters, and aggregates data into final clusters using an improved graph-cut technique based on path-based similarity.

**Claims And Evidence:**

TANGO addresses the issue that current mode-seeking methods identify modes by breaking certain dependency connections but rely heavily on local data characteristics, requiring case-by-case threshold settings or human intervention to be effective for different datasets. Experimental results demonstrate the effectiveness of this approach.

**Essential References Not Discussed:**

None.

**Experimental Designs Or Analyses:**

The experimental results are comprehensive.

**Methods And Evaluation Criteria:**

TANGO introduces a novel evaluation metric for computing the typicality of a point and employs an improved spectral clustering technique to aggregate typical subclusters.

**Other Comments Or Suggestions:**

The authors are encouraged to carefully address the points raised in the Weaknesses section and provide strong responses.

**Other Strengths And Weaknesses:**

Strengths:
1. The authors integrate both local and global distribution characteristics of data and propose a novel clustering framework that fuses local and global information, introducing a new global perspective into mode-seeking methods.
2. By introducing typicality, the authors reveal the global significance of data points under locally defined density-based dependencies and use typicality to detect modes in a fully automated manner. Additionally, they provide theoretical analysis and an efficient method for computing typicality.
3. The authors design an improved path-based similarity method to comprehensively and effectively assess the similarity of subclusters and adopt a graph-cut method to determine the final clustering.

Weaknesses:
1. Section 4.4: The authors claim to aggregate subclusters using the graph-cut method after obtaining the modes and corresponding tree-like subclusters based on typicality. However, further clarification is needed on how the aggregation operation is performed.
2. Section 4.4: The authors apply spectral clustering to tree-like subclusters based on path-based similarity to obtain the final clustering results. Since it is well known that spectral clustering requires a specified number of clusters, the method could potentially lead to a trivial solution where all data points are assigned to a single cluster if the number of clusters is not pre-specified. The authors should address this concern and clarify how this issue is resolved.
3. Appendix C.6 (Ablation Study): The authors should further clarify the differences between TANGO-a and TANGO-b and whether the experimental settings are identical. Additionally, it is recommended that they include an ablation study examining the effect of using only the Typicality-Aware Mode-Seeking technique without applying the Aggregating Mode-Centered Subclusters technique.
4. Minor textual errors: Line 350 contains a redundancy: "clustering labels labels."

**Questions For Authors:**

None.

**Relation To Broader Scientific Literature:**

Based on the challenges in existing work, the authors propose a novel approach.

**Theoretical Claims:**

The paper presents an innovative density-based algorithm, but provides limited discussion on the theory.

---

> ### Author Rebuttal · Authors · 2025-03-30
>
> Thank you so much for reviewing our paper. We answer your questions below.
>
> Weakness 1: The aggregation operation is done by considering each tree-like subcluster as a vertex in a similarity graph, where the similarity between these subclusters is determined by a path-based connectivity, and finally the spectral clustering (specifically, NCut method) is applied to aggregate these vertices into a final partition with the specified number of clusters. More formal clarification on path-based similarity between subclusters is included in the proof of Theorem 3 in the appendix. We will further clarify this operation in the revised version.
>
> Weakness 2: In our experiments, we have pre-specified the target number of clusters for spectral clustering as the number of ground-truth clusters for each dataset, and fixed the target number of clusters at $5$ on image segmentation task. It is a common topic when the target number of clusters is not pre-specified in spectral clustering, and there exist many classical methods such as Eigengap-based method, Modularity-based method and Self-Tuning Spectral Clustering method to deal with this situation. We can also integrate these methods to automatically determine the target number of clusters when it is not pre-specified.
>
> Weakness 3: We will make the differences between TANGO-a and TANGO-b clearer. In TANGO-a, we removed the whole typicality-aware mode-seeking step and just directly apply spectral clustering on all points in dataset with path-based similarity. This is to show that the typicality-aware mode-seeking step is essential. In TANGO-b, we include the mode-seeking step but not typicality-aware, to further validate the significance of typicality. We have conducted the ablation study you suggested by removing the aggregating component to validate its necessity, and observed significant performance decrements (ARI dropped from 39.4, 64.21, 65.37, 63.49, 39.77, 49.15, 59.57, 82.88 to 13.96, 39.44, 36.85, 36.42, 23.16, 6.01, 47.24, 37.19, resp.) on 8 datasets used in current ablation study. We will include this discussion in the revision.
>
> Weakness 4: Sorry for the confusion. In fact, the second "labels" is the name of the variable and "clustering labels" is a description of it. We will make this clearer.

---

### Decision · Program_Chairs · 2025-05-01

**Decision:**

Accept (poster)

**Comment:**

The paper proposes a global perspective metric (typicality) to quantify the confidence of a point being a mode, and designed an algorithm to compute typicality and provide theoretical analysis. Most reviewers have positive evaluation of the paper and, after reading the author's response, I believe that the downsides listed by the reviewer with the lowest score can be easily addressed.

Based on this and the overall positive evaluation of the paper, I recommend accept.